# The global distribution and drivers of wood density and their impact on forest carbon stocks

The density of wood is a key indicator of the carbon investment strategies of trees, impacting productivity and carbon storage. Despite its importance, the global variation in wood density and its environmental controls remain poorly understood, preventing accurate predictions of global forest carbon stocks. Here we analyse information from 1.1 million forest inventory plots alongside wood density data from 10,703 tree species to create a spatially explicit understanding of the global wood density distribution and its drivers. Our findings reveal a pronounced latitudinal gradient, with wood in tropical forests being up to 30% denser than that in boreal forests. In both angiosperms and gymnosperms, hydrothermal conditions represented by annual mean temperature and soil moisture emerged as the primary factors influencing the variation in wood density globally. This indicates similar environmental filters and evolutionary adaptations among distinct plant groups, underscoring the essential role of abiotic factors in determining wood density in forest ecosystems. Additionally, our study highlights the prominent role of disturbance, such as human modification and fire risk, in influencing wood density at more local scales. Factoring in the spatial variation of wood density notably changes the estimates of forest carbon stocks, leading to differences of up to 21% within biomes. Therefore, our research contributes to a deeper understanding of terrestrial biomass distribution and how environmental changes and disturbances impact forest ecosystems.

Wood density, defined as the dry mass per fresh volume of wood, is a fundamental functional trait which reflects the carbon investment of trees. It is closely linked to the life history and functional attributes of trees, including mechanical and physiological properties[1–3]. Wood density plays a crucial role in determining the competitive ability of tree species and shapes the composition, structure and function of forest ecosystems[4–7]. These dynamics affect the rate of tree mortality[8] and wood decomposition[1], which are central to how ecosystems respond to environmental changes. Furthermore, the strong link between wood density and biomass production[1,9] makes it a vital factor in quantifying

terrestrial carbon uptake and storage[10–13]. Over one-third of the total variation in aboveground biomass in tropical forests can be explained by spatial differences in wood density[1,14]. Yet, until now, we lack a spatially continuous understanding of the variation in wood density in angiosperms and gymnosperms that would be necessary for representing this information in global forest carbon storage estimates.

In recent decades, empirical and theoretical studies have identified a wide range of factors that shape global variation in tree wood densities, including abiotic variation, biotic conditions, successional stages and human disturbances[1,9,10,15–18]. The evolution of wood density

✉e-mail: lidong.mo@usys.ethz.ch

**Fig. 1 | Observed wood densities across the global forest inventory plots and within gymnosperms, angiosperms, forest types and biomes.**
**a–c**, Wood density distribution of gymnosperm (**a**) and angiosperm (**c**) species and influence of the proportion of angiosperms on CWD (**b**). The wood density distribution in gymnospermous species is narrower and has a smaller mean (~20% lower) than in angiospermous species. **b**, CWD increases with increasing proportion of angiospermous species in forest communities. We included

8,249 taxa with information on angiosperms and gymnosperms comprising 8,036 angiosperms and 213 gymnosperms, each with wood density information available at the species or genus level. **d**, Map of CWD observations for the ~1.1 million plots from the GFBi database. **e,f**, Box plots of observed CWD at the forest type (**e**) or biome level (**f**). Box plot shows the median, interquartile range and whiskers for data spread, excluding outliers.

is fundamentally shaped by the cost for wood construction and the need for biomechanical and hydraulic safety[2,19,20]. Denser wood offers enhanced mechanical support and greater resistance to drought conditions in the xylem but this advantage may be offset by the higher resource allocation required for wood production, resources that could otherwise support growth or reproduction[21–23]. Consequently, in ecosystems with higher vapour pressure deficits, such as warm and dry forests, trees are likely to develop denser wood to maintain xylem resistance against implosion and rupture[21,23]. In contrast, in warm and humid ecosystems with lower vapour pressure deficit, life history strategies may lean towards rapid growth, characterized by reduced carbon investment in wood, to maximize competitive ability[21,22]. In

colder regions, gymnosperms with low-density tracheids have a competitive advantage over angiosperms. Tracheids of gymnosperm trees, being narrower than the cavitation threshold of 30 μm, are capable of functioning under water and freezing stress, which allows them to resume transpiration early in the spring[24,25]. Additionally, factors such as reduced canopy height or a lower prevalence of pathogens[26] in colder regions may reduce the need for high investment in wood construction[27]. As a result, the balance between the investment of trees in wood construction and their mechanical and physiological safety is expected to lead to notable geographic variations in wood density worldwide, affecting the structure, function and diversity of ecosystems.

Wood density also varies with the successional stage of forests[28] and is influenced by disturbances from both natural processes and human activities[1,29–36], such as wildfires[17,37–39]. For example, in parts of the Amazon rainforest, wood density in secondary forests was found to be 33% lower compared to predisturbance conditions[40,41]. This reduction is attributed to the prevalence of early-successional species with less dense wood in disturbed tropical wet and moist forests[1,14,17,33,37,40–45]. Conversely, in tropical dry forests, wood density often increases postdisturbance as a result of the establishment of more conservative, slow-growing species which are resistant to environmental stresses[9,46–50]. This implies that forest wood density responds unevenly to disturbances under different environmental conditions[33,51]. Yet, such context-dependency remains untested at a global scale. Understanding the global distribution of forest wood density and the various influencing factors, including climate and ecosystem disturbances, is vital for predicting and managing the responses of forest ecosystems to environmental shifts and for formulating effective strategies to mitigate and adapt to climate change impacts.

Here we paired -1.1 million ground-sourced forest inventory plots (Fig. 1d) from the global forest biodiversity initiative (GFBi) database[52] with collated species-level wood density data[1,53–60] to explore global variation in wood density among both angiosperm and gymnosperm trees. Using this large-scale observation approach, we tested competing hypotheses about the dominant factors driving wood density variation across global forests, including temperature, water availability, species composition and disturbances. This approach allowed us to test theoretical predictions of geographic variation and to create a global model of wood density (Fig. 1 and Methods). We calculated community-wide mean wood density (CWD) by weighting the wood density of each individual observed in a forest plot by its basal area. To explore responses to anthropogenic and natural disturbance gradients, we integrated our observations with global information on human disturbance[61] and fire frequency[62]. Finally, we estimated the total live forest biomass by integrating our CWD map with spatially explicit data on live tree volume[63,64], root mass fraction[65] and biome-level biomass expansion factors (Supplementary Table 1).

## Spatial and phylogenetic wood density variation

Gymnosperm trees, which are dominant in boreal and high elevation regions, had 20% lower wood density than angiosperms, with mean densities of $0.47 \pm 0.07$ g cm$^{-3}$ and $0.59 \pm 0.14$ g cm$^{-3}$, respectively. Accordingly, the CWDs of the global forests were positively related to the proportion of angiosperms within a plot (Fig. 1b).

Our global CWD data reveal strong differences in wood density across the major forest regions ('Plot-level wood density metrics' in Methods). Compared to boreal regions, which have a mean CWD of $0.46 \pm 0.05$ g cm$^{-3}$, the average CWDs in temperate ($0.52 \pm 0.09$ g cm$^{-3}$; mean $\pm$ s.d.), tropical ($0.57 \pm 0.10$ g cm$^{-3}$) and dryland ($0.59 \pm 0.09$ g cm$^{-3}$) regions were 13%, 24% and 28% higher, respectively (Fig. 1e and Supplementary Table 2). At the biome level, tropical coniferous and Mediterranean forests had the densest wood, each with a wood density of 0.6 g cm$^{-3}$. The standard deviations are $\pm 0.14$ g cm$^{-3}$ and $\pm 0.09$ g cm$^{-3}$, respectively. The lowest wood densities were observed in boreal ($0.46 \pm 0.05$ g cm$^{-3}$) and temperate ($0.49 \pm 0.07$ g cm$^{-3}$) coniferous forests and flooded savanna ($0.46 \pm 0.08$ g cm$^{-3}$) regions, with densities 23% to 32% lower than in tropical coniferous and Mediterranean forests (Fig. 1f and Supplementary Table 3). There was also considerable variation in CWD within biomes, which can rival the amount of variation across biomes.

To examine how phylogenetic position affects wood density variation across different species, we used a dated phylogeny on 4,298 species in 189 families and 55 orders. We found a pronounced phylogenetic signal, supporting niche conservatism in wood density among these evolutionary distinct lineages (Pagel's lambda = 0.92, $P < 0.01$ and Blomberg's $K = 0.01$, $P < 0.01$)[66,67]. Similarly, ref. 68 reported a lambda

value of 0.77 using wood density information from 2,261 species worldwide. This evolutionary signal persists at the order level[69], indicating that higher wood densities in the angiosperm orders Myrtales (0.74 g cm$^{-3}$), Fabales (0.69 g cm$^{-3}$), Ericales (0.68 g cm$^{-3}$) and Fagales (0.64 g cm$^{-3}$) and lower wood densities in the Pinales (0.45 g cm$^{-3}$), Cupressales (0.50 g cm$^{-3}$), Araucariales (0.50 g cm$^{-3}$), Malvales (0.50 g cm$^{-3}$), Rosales (0.53 g cm$^{-3}$) and Laurales (0.54 g cm$^{-3}$) are phylogenetically conserved over evolutionary time (Fig. 2).

## Geospatial mapping

To map the geographic variation of wood density based on its relationship with environmental factors, we developed random-forest models using 62 global layers of climate, topography, soil, vegetation and human activity (Supplementary Table 4). These models were applied to all tree species (Fig. 3a), as well as separately to angiosperms (Fig. 3b) and gymnosperms (Fig. 3c). We observed spatial autocorrelation in model residuals[70] up to a distance of 50 km (Supplementary Fig. 1). To mitigate the effect of spatial autocorrelation and ensure the reliability of our model predictions, we used a spatial bootstrapping procedure: we created 200 bootstrapped training subsets, each with data points at least 50 km apart (Methods). We then built individual models for each subset. Our final model, with 62 predictors, achieved a global average $R^2$ of 0.53 (tenfold cross-validation; Supplementary Fig. 2). This model was used to map global wood density trends, revealing lower densities at higher latitudes and elevations (Fig. 3). For example, forests in Canada, Siberia, the Alps and the Qinghai-Tibetan plateau showed low wood density (<0.5 g cm$^{-3}$), whereas high-density areas (>0.6 g cm$^{-3}$) included warm, arid regions like the African Savanna and Australian open forests.

To assess the predictive uncertainty of our models, we calculated the bootstrapped coefficients of variation (standard deviation divided by mean) for CWD values. These results showed high confidence in predictions across all models, with coefficients of variation <5% for all pixels in existing forest areas (Supplementary Fig. 3). Furthermore, we distinguished between model interpolation (predictions within the environmental range of the training data) and extrapolation (predictions outside this range) using a principal component analysis (PCA)-based approach. Our analysis indicated that >95% of the forested areas fell within the environmental range of our training data in >95% of cases. Most of the outliers were located in African savanna regions, probably due to lower sampling density in these regions (Supplementary Fig. 4).

## Drivers of global wood density variation

To assess the relative importance of climatic, soil, vegetation and disturbance factors in driving global CWD, we used partial regression and random-forest modelling (Fig. 4). We selected nine variables including environmental factors and functional traits based on previous research[1], including mean annual temperature, soil moisture, soil carbon-to-nitrogen (C:N) ratio (indicating nitrogen availability[71]), leaf area index (LAI; indicating growth and canopy light competition[72]), tree diversity (species richness), forest age, diameter at breast height (DBH), human modification and fire frequency. Differences in the relative occurrence of angiosperms versus gymnosperms were accounted for by including the plot-level angiosperm ratio as an additional predictor (Fig. 4a,b). Overall, this analysis revealed that mean annual temperature is the most influential factor on CWD. Specifically, a 1 °C increase in temperature correlates with an average 0.5% increase in wood density (Supplementary Table 5). This trend was consistent in separate analyses of angiosperm and gymnosperm communities (Fig. 4c–f) and across forest types and biomes (Fig. 4d,f). The effect of water availability, nutrient resources and temperature on CWD is in alignment with the study conducted by ref. 73, which used soil water-holding capacity, soil basicity index and elevation as proxies for these factors.

The relationships between other tested variables and CWD varied considerably across forest types. High soil moisture correlated with

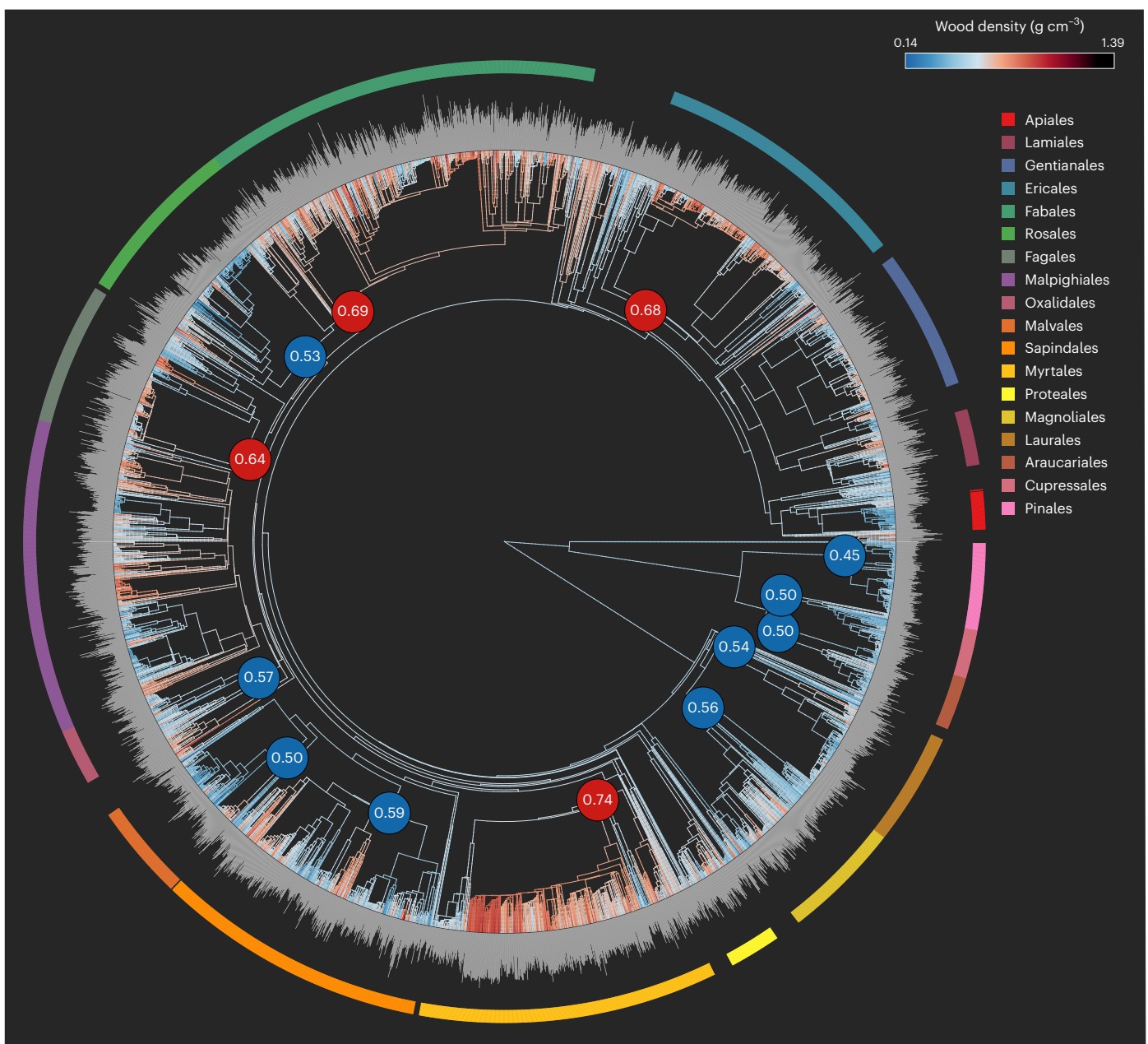

**Fig. 2 | Phylogenetic tree and wood density information of global tree species.** The phylogenetic tree was constructed using the R package V.PhyloMaker, with wood density information available for 4,298 species (189 families from 55 orders). Wood density exhibits a strong phylogenetic signal (Pagel's lambda = 0.92, $P < 0.01$, Blomberg's $K = 0.01$, $P < 0.01$). The colours of the branches and the grey bars at the tips represent the wood density of each species. To identify orders that have significantly different wood densities compared to all other tree species, we conducted a two-tailed significance test by comparing the order-level wood density with 999 randomized wood density values from the entire phylogenetic tree. The coloured circle surrounding the phylogeny represents different orders. The filled blue/red circles inside the phylogeny indicate orders that show significantly ($P < 0.05$) lower (blue) or higher (red) wood densities relative to all the species. Numbers inside the circles represent the average wood density of the respective order.

low CWD in tropical and temperate forests but led to higher CWD in boreal and dryland forests (Supplementary Fig. 5). In tropical regions, LAI was positively correlated with CWD, whereas a negative correlation was observed in temperate, boreal and dryland regions. The soil C:N ratio generally correlated with slight decreases in CWD but an inverse relationship was observed in boreal forests. Forest age, while less influential on a global scale, displayed negative effects across all forest types (Supplementary Fig. 5). This pattern reflects the consistent impacts of forest age in communities dominated by angiosperms and gymnosperms (Fig. 4). Plot-level mean DBH, which may also reflect forest age to some extent, had a minor impact on wood density globally (Fig. 4a).

The impact of major disturbances, specifically human activity and fire frequency, on CWD was highly context-dependent. Our analysis across all plots showed human modification as the third most important factor affecting CWD (Fig. 4a) but its importance diminished in gymnosperm-only communities (Fig. 4c,e). This suggests that human activities indirectly influence CWD, primarily by altering the proportion of coniferous and broadleaved trees. Fire frequency was the least impactful factor among the nine variables (Fig. 4a). The limited global effect of fires is probably due to their infrequent occurrence in forest worldwide, with 96% of global forests not experiencing fires in the past 20 years[62]. However, the long-term impacts of fires on forest composition may be underestimated in our analysis, as their influence

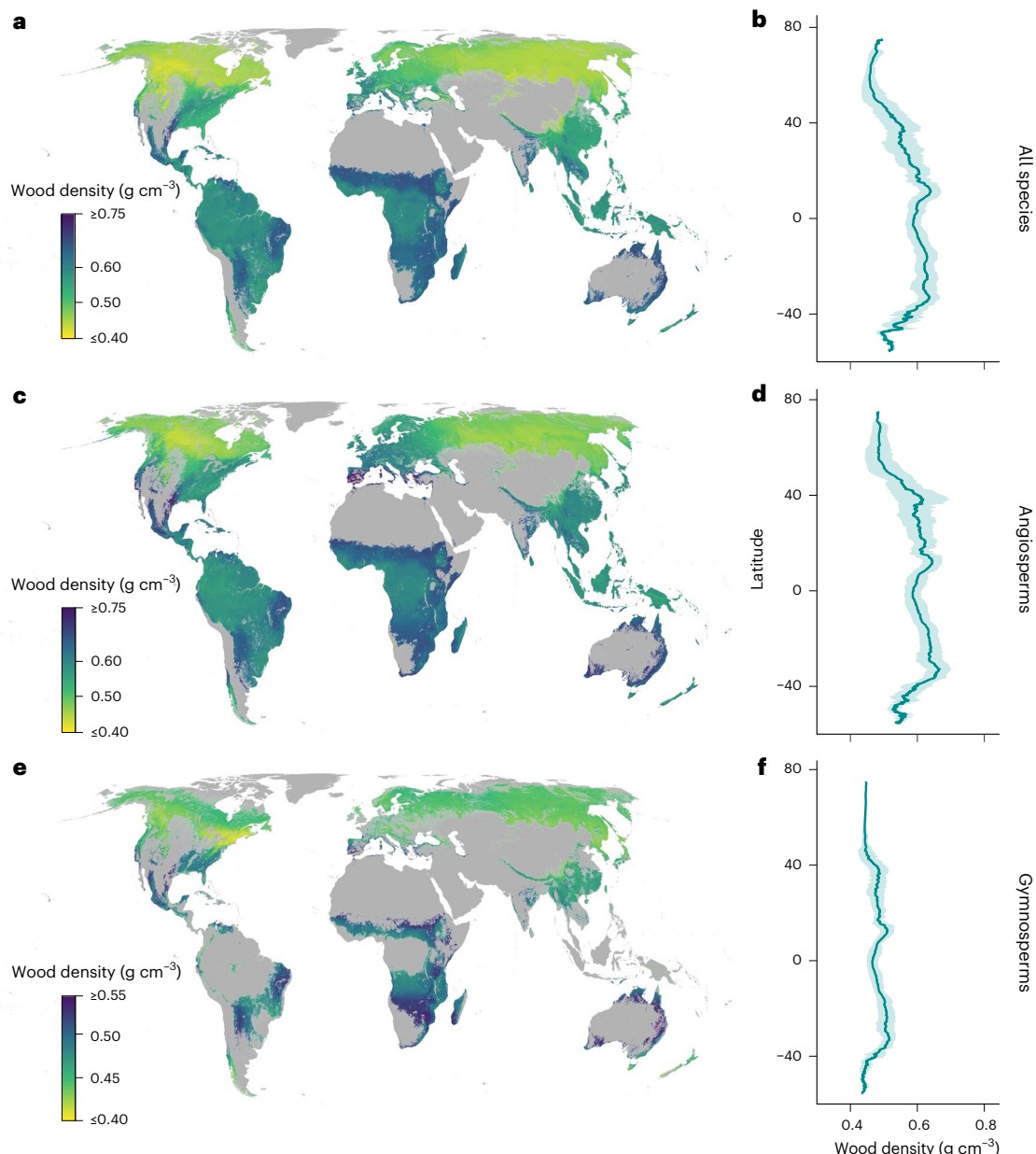

**Fig. 3 | Global maps of wood density. a,c,e,** Wood density maps for all species (**a**), angiosperms-only (**c**) and gymnosperms-only (**e**). **a**, The community-level wood density map was derived from an ensemble approach, averaging the global predictions from the 200 best random-forest models. **c,e,** Angiosperm-only (**c**) and gymnosperm-only (**e**) wood density maps were derived from ensemble averaging of the global predictions from the 100 best random-forest models, respectively. **b,d,f,** Corresponding latitudinal trends in wood density aggregated for each 0.1 arc degree latitude: all species (**b**), angiosperms (**d**) and gymnosperms (**f**). Error ranges represent 1 s.d. either side of the mean. Maps are projected at 30 arcsec (~1 km²) resolution. Non-forested areas are displayed in grey. In the wood density maps for angiosperms (**c**) and gymnosperms (**e**), we correspondingly excluded pixels where angiosperms and gymnosperms constituted <5% of the entire community.

can extend beyond the 20 year period we considered. Additionally, the intensity of fire, a crucial aspect of fire disturbance[74], was not captured in our frequency data, probably explaining why we did not find a stronger effect of fire frequency[74].

To further explore how environmental variables modulate the relationships between disturbance processes and CWD, we conducted recursive partitioning analyses. These analyses show that in cold regions (<10 °C), CWD increases with human disturbance, whereas in warmer areas, it decreases (Supplementary Fig. 6a). Similarly, the effect of fire frequency on CWD also varied with temperature: it slightly reduces CWD in colder climates but increases it in warmer ones (Supplementary Fig. 6b). The relationships between disturbances and CWD

were also dependent on the proportion of angiosperms versus gymnosperms in a forest. These findings underscore the context-specific nature of the effects of human disturbance and fire frequency on wood density, influenced by factors such as temperature and forest taxonomic composition.

## Wood density and global biomass estimates

To assess the impact of wood density variations on global forest biomass estimates, we integrated our wood density map with the latest global maps of live tree volume[63], root mass fraction[65] and biomass expansion factors[75] (Fig. 5a and Supplementary Table 1). This analysis revealed a total tree biomass of 374 GtC (Supplementary Fig. 7), of which 200 GtC

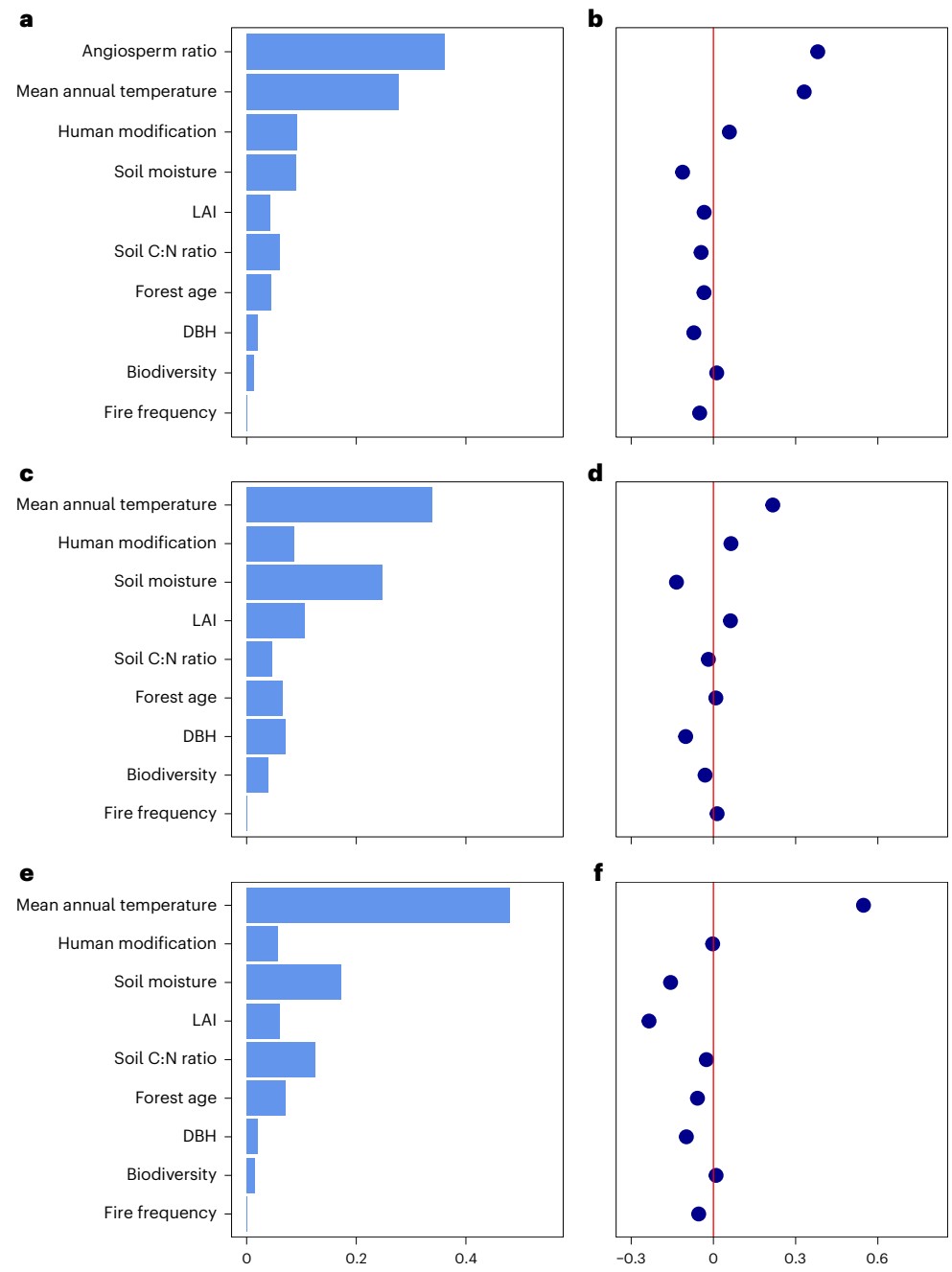

**Fig. 4 | Variable importance of the selected environmental metrics. a–f,** The environmental metrics are based on random-forest models (**a,c,e**) and linear partial regression models (**b,d,f**). **a,b,** Variable importance of the selected covariates across global forests, including angiosperm ratio to control for wood density differences between angiosperms and gymnosperms. **c,d,** Variable importance within angiosperm-only communities. **e,f,** Variable importance within gymnosperm-only communities. Mean decrease in accuracy values in **a,** **c** and **e** represents the relative contribution of each variable to CWD variation, whereby we averaged the values of 100 bootstrapped random-forest models. Bootstrapped partial regression coefficients for each variable (**b,d,f**) were calculated by averaging the partial regression coefficients from 100 multivariate linear models. All variables were standardized to allow for direct effect size comparison. In addition, we quantified the absolute effects of these covariates using partial regression analysis, as detailed in Supplementary Table 5.

(53.3%) is stored in tree stems, 93 GtC (24.9%) in branches, foliage and other aboveground living parts and 81 GtC (21.7%) belowground as roots. This global estimate aligns well with previous estimates based on remote sensing, ground-sourced models or harmonized ensemble approaches[63,76,77], estimating total tree biomass in the range of 354–445 GtC (Supplementary Fig. 7). However, our wood density-based biomass estimations present spatial deviations compared to previous studies, showing an agreement in spatial variation ranging from 45% to 93% with earlier research[13,63,76–78] (Supplementary Fig. 7a). Our estimates

were most closely aligned (93%) with those from GlobBiomass[63], as both used the same live tree volume data (Supplementary Fig. 7c).

To isolate the influence of wood density variation on global tree biomass distribution, we compared our wood density-informed biomass model with a model using a constant wood density value of 0.53 g cm⁻³ (the global average). We found that the constant wood density model estimated the global biomass to be about 4% lower than the spatially explicit wood density model (359 GtC compared to 374 GtC; Fig. 5b). However, significant differences emerged within

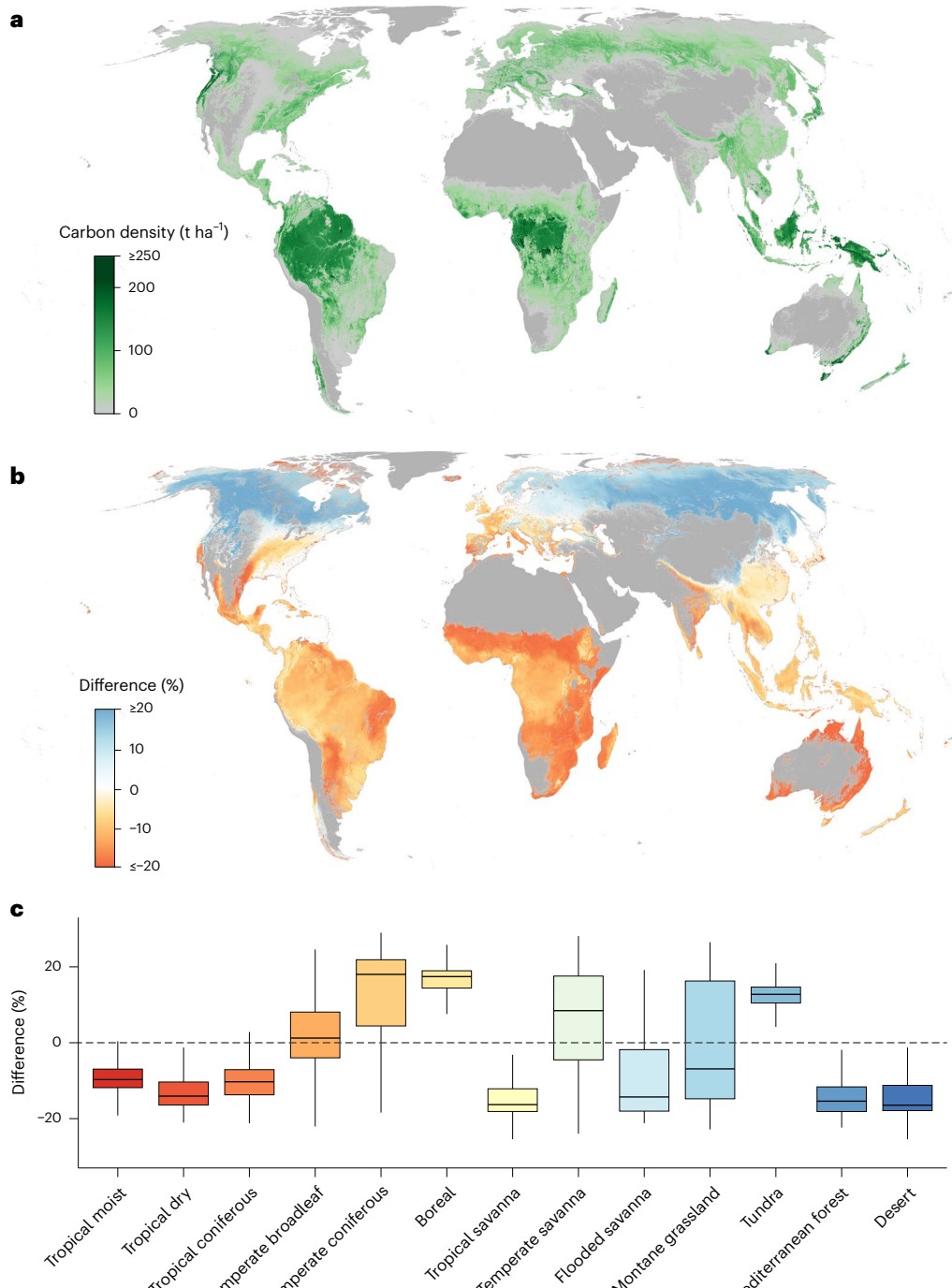

**Fig. 5 | Comparison of global living tree biomass distribution using spatially explicit wood density data versus a universal wood density value. a**, The global distribution of living tree biomass (in tonnes per hectare), derived by integrating our wood density map with spatially explicit data on living tree volume, root mass fraction and biomass expansion factors. **b**, Percentage difference in estimated living tree biomass when comparing results derived using the global wood density map (from **a**) with estimates using a single, universal wood density value. The difference is calculated as the percentage change by subtracting the spatially explicit estimate from the universal estimate and then dividing by the spatially explicit estimate. Blue areas show regions where the universal estimate is higher, and red/orange areas indicate where the spatial estimate is higher. **c**, Percentage difference between the two biomass estimation methods across biomes. Box plots show the median, interquartile range and whiskers for data spread, excluding outliers.

various biomes (Fig. 5b), where the constant density model underestimated carbon stocks in specific biomes like tropical moist, tropical dry, tropical savanna and Mediterranean forests by, on average, 12%, 17%, 17% and 21%, respectively, and overestimated them in temperate coniferous and boreal forests by 10% and 13%.

These findings underscore the critical role of spatially explicit wood density estimates in accurately predicting forest carbon stocks,

taking into account the variations across different regions and forest types. Our detailed wood density map therefore allows for a more accurate representation of the geographic variation in tree carbon storage.

## Discussion

Owing to high heterogeneity in wood density across forest types[79,80] and along successional stages, modelling the spatial variation in forest

functional traits has remained a major research challenge[16,79,80]. Yet, recent advancements in big data and remote sensing tools have begun to provide detailed high-resolution information on environmental, human disturbance and vegetation characteristics. In this study, we integrated these features to model global variations in wood density and identify the factors driving this variation.

Our analysis identifies temperature as the dominant driver of global wood density variation, with a more than three times greater effect than any other variable (Fig. 4a), highlighting temperature as a selective pressure and filter of global wood density[2]. Other factors, such as soil moisture, also affect wood densities but are more noticeable at regional scales. In addition, mean annual temperature is highly correlated with evapotranspiration and potential evapotranspiration (Pearson's correlations around 0.9). Warmer climates thus often coincide with significant water stress. These global-scale trends support the hypothesis that the need for hydraulic safety drives high wood densities in stressful environments[2,19,20].

The observed positive correlation between community-level wood density and temperature[2,10] leads to distinct latitudinal and elevational trends (Fig. 3a,b). While changes in gymnosperm versus angiosperm species composition amplified these trends, the environmental factors driving global wood density variation were remarkably similar among these distantly related groups (Fig. 3c,e). Temperature and water availability had a consistent effect on wood density across angiosperms and gymnosperms (Fig. 4c,e), indicating similar wood anatomical adaptations to environmental factors.

Our findings on the drivers of global forest wood density align with ref. 81, which highlighted the importance of leaf habit, temperature seasonality, cloud cover and annual precipitation. This study[81] calculated pixel-level average wood density by averaging individual tree-level measurements within pixels. In contrast, we integrated forest inventory data to represent CWD. Overall, our wood density estimates show slightly greater variation across the globe. We estimate higher densities than ref. 81 in warm tropical regions and lower densities in cold boreal regions. At the biome level, our predictions are higher in denser wood biomes and lower in regions with low wood density. Despite these differences, both studies show similar patterns of wood density across global forests, with higher wood densities at lower latitudes and lower densities at higher latitudes. The overall $R^2$ between the two models is 0.58 (Supplementary Fig. 9). This consistency in global wood density patterns and climate responses is crucial for enhancing our confidence in predicting the impact of climate change on global forest biomass distribution and shifts in forest composition.

The importance of balancing hydraulic safety and growth efficiency has been well-documented in tropical dry forests[82]. Indeed, our model predicts the highest wood densities for hot environments with low water availability, such as the dryland regions of South America, Africa and Australia. Conversely, in tropical moist forests, wood density correlates more strongly with growth and mortality rates than with resistance to cavitation. This is evident in pioneer trees, which have low wood density and high growth rates, whereas non-pioneer rainforest trees tend to have higher wood density, largely due to their longevity and competitive advantage, rather than drought stress[80]. Overall, wood density serves as a multifaceted proxy for environmental interactions and responses, as demonstrated by the divergent trends in community wood density observed in dry versus wet tropical forests[9].

The negative correlation between soil moisture and wood density lends support to observations in previous local-scale analyses[1,2,83,84], a trend that can largely be explained by the high abundance of slow-growing species in dry environments[9,50,85–87]. These species typically adopt a conservative resource-uptake strategy and exhibit high water-use efficiency. Previous studies have indicated variability in the relationship between wood density and water availability, with differences in the direction and magnitude depending on the research scale and species sampled[2,84,88]. Our global-scale sampling now shows that

dry forests have wood densities up to 31% higher than those found in more humid regions[89] (Figs. 1e and 3). Nevertheless, we also find regional differences in the effect of soil moisture (Supplementary Fig. 5). For example, a positive relationship between wood density and water availability was found within boreal forests, while the opposite was true for tropical and temperate regions. Such variation in the effect of soil moisture may be driven by variation in functional properties of species. For instance, water availability tends to be less limiting in broadleaved trees than in conifers because angiosperm vessels can be more efficient in water conduction than tracheids[24].

Although biodiversity (species richness), forest age, DBH, LAI and soil C:N ratio are important factors influencing local variation in wood density, these effects were overwhelmed by the impacts of temperature, soil moisture and angiosperm ratio at a global scale. This may suggest that biotic interactions play a relatively smaller role in shaping broad-scale variation in wood density, relative to abiotic environmental factors. Within gymnosperms, we observed a negative correlation between C:N ratio and wood density. This might reflect an increased investment in xylem safety in nutrient-rich environments, where the construction of tissues (carbon acquisition) is less of a limiting factor[90]. Forest age had opposite effects on wood density variation in communities dominated by angiosperms and gymnosperms. This indicates divergent trajectories in wood density development during the maturation of these two types of forests (Fig. 4b,c).

The relationship between wood density and disturbance was highly context-dependent, with regional contingencies being dependent on the environmental background conditions of the region[9,17,33,37,41,45–47,50]. While previous studies have emphasized the role of water availability[51], our recursive partitioning analyses suggest mean annual temperature as the main driver shaping the relationship between wood density and forest disturbances, such as human disturbance and fire frequency (Supplementary Fig. 6). This might be driven by regional differences in the relative trade-off between acquisitive and conservative resource-uptake strategies[9,50,85–87]. In tropical wet forests, seral communities on disturbed forest margins are often dominated by short-lived, light-demanding species which tend to have low wood densities[41,91], while in tropical dry forests, seral communities often consist of drought-resistant species with dense wood[91–93]. Controlled experiments and long-term field observations will be needed to further disentangle the context-dependent responses of wood density to disturbances.

Our analysis highlights the strong role of species composition in shaping wood density variation[1,2,84,94–96]. Owing to the strong phylogenetic signal in wood density, related species display similar wood densities, even when growing under different environmental conditions[84]. For example, gymnosperm tree species have, on average, 20% lower wood densities than angiosperm species (Fig. 2). The anatomical differences between angiosperms and gymnosperms play a crucial role in this disparity. Gymnosperms are characterized by thinner cell walls and smaller pits, decreasing the risk of xylem cavitation at the expense of hydraulic conductivity. In humid, hydraulically less stressful environments, angiosperm trees with higher hydraulic conductivity might thus tend to outcompete gymnosperm trees[2]. This may add to the observed decreases in wood density from low to high latitudes (Fig. 3a) or from tropical to boreal forests (Fig. 3b)[1,2]. Interestingly, we find consistent biogeographical trends in wood density for gymnosperms and angiosperms, with temperature being the key regulator. This is particularly evident in the pronounced increase in wood density from boreal to tropical forests within gymnosperm species (Fig. 3f). This trend illustrates the strong selective pressures and filters of temperature on tree wood density patterns globally (Fig. 4a,b).

## Conclusion

The integration of global ground-sourced forest inventory data with wood density measurements allowed us to quantitatively assess the environmental factors driving the wood density distribution on a global

scale. This integration has resulted in a high-resolution global model, providing critical information on the structure and biomass distribution of the forests of the world. Our analysis identifies taxonomic composition—particularly the distinction between angiosperms and gymnosperms—as the primary biotic driver influencing global wood density variations. Temperature, in conjunction with water availability, emerges as the dominant abiotic factor that controls the global variation in wood density. This pattern is probably attributable to the role of denser wood in enhancing competitive ability, hydraulic efficiency and transpiration efficiency in warmer environments. We also observed that community-level wood density responses to disturbances vary across forest types, biomes and environmental conditions. By integrating our wood density map with other key metrics such as live tree volume[63], biomass expansion factors and root mass fractions[65], we could benchmark existing forest biomass stocks, estimating a total living biomass of 374 GtC. Our research also showed that biomass estimates within biomes could vary by as much as 21%, depending on whether the variability of wood density was considered or if wood density was assumed to be uniform worldwide. Our findings contribute to an improved understanding of the structure and biomass distribution in global forests and highlight the effects of human and environmental disturbances on global forest communities and functional traits.

## Methods

### Data sources

Wood density is commonly measured as the ratio of the oven-dry mass of a wood sample to its green volume. Most wood density data stem from wood core samples[59], while some are derived from fresh volumes and dry weights of whole tree components[58,60]. We compiled wood density measurements of individual trees or aggregated to the species level from several databases or publications. The majority of observations came from the Global Wood Density Database by refs. 1,57, encompassing a total of 16,468 records in 8,412 species and the TRY database[59] with a total of 46,668 records for 7,514 species. Additionally, 1,117 wood density records in 937 species came from ref. 53, 4,022 records in 872 tree species from ref. 55, 618 records in 615 species from ref. 56, 624 records in 250 species from ref. 60, 3,529 records in 179 tree species from ref. 54, 3,092 records in 58 species from ref. 58 and 1,234 records in 1,061 species from published research articles by searching for 'wood density' in Google Scholar (publications listed in Supplementary Data 1). After standardizing the taxonomic names using the Taxonomic Name Resolution Service[97] (R package TNRS v.0.1.0) and removing synonyms, we obtained 77,372 wood density observations across 10,703 species and 2,026 genera (data are available at GitHub https://github.com/LidongMo/GlobalWoodDensityProject).

To test the compatibility in wood density estimates among databases, we conducted an analysis to quantify their similarity. By constructing a linear regression model based on common species pairs, we calculated an $R^2$ value of 0.78, indicating high consistency among all nine data sources and minimal bias introduced by different wood density determination methods (Supplementary Figs. 8 and 10).

### Phylogenetic and trait analysis

To test whether wood density is phylogenetically conserved, we computed common phylogenetic metrics as well as random-effects models including taxonomic information. We built a phylogenetic tree using the R package V.PhyloMaker[98], with a total of 4,298 species (189 families from 55 orders) with wood density information matching the species in the phylogenetic database. To test for phylogenetic signal in wood density, we computed Pagel's lambda and Blomberg's $K$, using the phylosig function in the R package phytools[99]. To further test for trait conservatism at the order level, we used the ph_aot function from the R package phylocomr[100]. Order-level wood density values were calculated by averaging across all descendent terminal taxa[69] and comparing the means with 999 trait value randomizations across the tips of the full phylogeny

to obtain significance estimates[101]. Only orders for which we had data on at least 50 species were tested. We further quantified the extent of within-species and within-genus variation in wood density by running a random-effects model on all 77,372 observations, including species and genus as random effects and wood density as response variable. The model showed that ~81% of the individual variation in wood density is explained by taxonomic information on family, genus and species levels, with 24% of the variation explained by family information, 30% by genus information and an additional 27% explained by species information.

### Generating species- and genus-level wood density information

To quantify wood density variation across the world's forests, we assigned species-level wood density values to individual tree observations from the GFBi. The GFBi database consists of 1,188,771 unique forest census plots, containing data for all tree individuals with a DBH > 5 cm. Each plot contains information on geospatial coordinates (latitude and longitude in decimal degrees), individual-level species binomial name and DBH, plot size (median plot size = 25 m²) and measurement year. For remeasured plots, we kept only the latest observation year for our analysis. Across all plots, the mean observation year was 2003. To assess the impact of the temporal changes of forest community on the community-level wood density, we applied a random-effects model to plots with time-series information. The model, including wood density as dependent variables and plot and year information as random effects, showed that variance in wood density was predominantly (97.9%) attributable to differences across plots, with only 0.2% due to variations across years.

We then used the binomial names to assign species-level wood density information to the individuals in the GFBi database. As for the wood density information, species binomials in the GFBi database were standardized using the TNRS[97]. For species with more than one wood density record, we used the average of all available records. If no wood density information was available at the species level or if the GFBi individual was only identified to the genus level, mean genus-level wood density values were used instead. Because of the strong phylogenetic signal in wood density values, these genus-level estimates introduce only little error compared to species-level wood densities[2,10,96,102]. In total, species-level wood density data could be matched to 4,428 species included in the GFBi database, while genus-level data were matched to 1,192 GFBi genera. We excluded plots representing 0.4% of the total, where <75% of the individuals had wood density information at either the species or genus level. Consequently, 1,183,070 plots were included in our geospatial analysis.

According to the GFBi inventory plots, the global average tree diameters of angiosperms and gymnosperms were similar, at 21.9 and 21.8 cm, respectively, with 95% quantile ranges of 5.6–56.5 cm for angiosperms and 6.4–56.4 cm for gymnosperms.

### Plot-level wood density metrics

We allocated species-level wood density to each individual tree in the GFBi plots and calculated the CWD. This approach is supported by the phylogenetic conservatism of wood density (Fig. 2) and the small impact of individual-level wood density variations on community-level estimates[73]. The average community-wide wood density for each plot CWD was calculated as the wood density of all tree individuals weighted by tree basal area:

$$\text{CWD} = \frac{\sum_{i=1}^{n}(\text{WD}_{\text{tree}} \times B_{\text{tree}})}{\sum_{i=1}^{n} B_{\text{tree}}} \tag{1}$$

where $\text{WD}_{\text{tree}}$ is the wood density of each tree and $B_{\text{tree}}$ is the basal area of each tree.

The spatial modelling of community-wide wood density properties was performed at 30 arcsec (~1 km²) resolution and we therefore aggregated CWD values within each 30 arcsec pixel by calculating the mean, resulting in 506,630 pixel-level observations for the modelling.

To quantify CWD within biomes and forest types, we used all observations in each biome or forest type. Biome and forest types were classified using the WWF Biome map[103]. Forests were divided into the four broad categories (tropical, temperate, boreal and dryland), with tropical regions including six biomes (tropical and subtropical moist broadleaf forest, tropical and subtropical dry broadleaf forest, tropical and subtropical coniferous forest, tropical and subtropical grassland, savanna and shrubland, flooded grassland and savanna and mangroves), temperate regions including four biomes (temperate broadleaf and mixed forest, conifer forest, temperate grassland, savanna and shrubland and montane grassland and shrubland), boreal regions including two biomes (boreal forest/taiga and tundra) and dryland including two biomes (Mediterranean forest, woodland and scrub and desert and xeric shrubland).

### Environmental and human disturbance covariates

We used 62 covariates, representing information on climate, topography, soil, vegetation characteristics, fire frequency and human disturbances, to test for the effects of environment and anthropogenic disturbance on the global variation in CWD and create spatially explicit models which allow us to interpolate CWD across the globe. All covariates were available as global layers at 30 arcsec resolution: layers for 19 bioclimatic variables came from the CHELSA open climate database (www.chelsa-climate.org)[104]; topographic information (elevation, roughness, slope, profile curvature, northness, eastness and topographic position index) from the EarthEnv database (www.earthenv.org/topography)[105]; cloud cover properties (annual mean, interannual standard deviation and intra-annual standard deviation) from the EarthEnv (www.earthenv.org/cloud) database and ref. 106; depth to the water table from ref. 107; the annual mean of solar radiation, wind speed and vapour pressure from the WorldClim database (v.2)[108]; absolute depth to bedrock and soil texture (clay content, coarse fragments, sand content, silt content and soil pH), averaged for soil depths from 0 to 100 cm below surface, from the soil grids database[109]; soil moisture was down-scaled from 10 km resolution maps sourced from GLDAS2.0 (ref. 110), ERA5 (ref. 111) and MERRA2 (ref. 112), soil nutrient information (cation exchange, C:N ratio and nitrogen) from the WISE30sec database[71] and soil grids[109]; normalized difference vegetation index, enhanced vegetation index (upscaled from 250 m resolution), FPAR, LAI (upscaled from 500 m resolution) and annual net primary productivity from MODIS data[113–115]; aridity index and potential evapotranspiration from refs. 116,117; and current forest tree cover, tree density, canopy height and forest age from refs. 118–121, respectively.

To represent human and natural disturbances in our model, we used eight global layers that directly reflect anthropogenic disturbances: cultivated and managed vegetation and urban built-up[122], agricultural land use (cropland, grazing, pasture and rangeland transformed to pixel-level percentages)[123,124], human modification, reflecting the intensity of human activity[61] and natural disturbances of forests: fire frequency[62]. Human modification is the most comprehensive and representative human activity variable integrating five major human disturbance categories: human settlement, agriculture, transportation, mining and energy production and electrical infrastructure[61]. The map of fire frequency was generated from yearly observations of fire occurrence[62], by calculating the proportion of years with fire in each 30 arc degree resolution pixel. All covariates were extracted via Google Earth Engine[125]. The eight disturbance variables were uniformly scaled to represent a continuous gradient of human activity or fire frequency, whereby values of 0 indicate no disturbances in the respective pixel and values of 1 indicate maximum disturbance.

### Representation of training data

To evaluate the extent of interpolation versus extrapolation in our models, that is, how well our training data represents the full multivariate environmental covariate space, we performed a PCA-based approach

following ref. 126. We projected the covariates composite into the same space using the centring values, scaling values and eigenvectors from the PCA of the training data. Then, we created convex hulls for each of the bivariate combinations from the top principal components (which collectively covered >90% of the sample space variation). We used 22 principal components with 231 combinations for all covariates. Using the coordinates of these convex hulls, we classified whether each pixel falls within or outside each of these convex hulls. This analysis revealed that 95.2% of land pixels excluding Antarctica are covering at least 95% of the environmental conditions present in our training data locations (Supplementary Fig. 4).

### Geospatial modelling of global forest wood density properties

To train spatially explicit CWD models across the world's forests, we ran a series of random-forest machine learning models. The models included 62 predictor variables representing climate, soil, topography, vegetation, fire frequency and human disturbances. Parameter tuning for each model was performed through the grid search function of the H2O R package[127] to iteratively explore the results of a suite of machine learning models trained on the 62 covariates.

To test for spatial autocorrelation of model residuals, we trained a generalized additive model (GAM) using the same 62 covariates and then extracted Moran's *I* values of the GAM residuals at spatial scales of 0–1,000 km. This analysis revealed positive spatial autocorrelation up to a distance of 50 km (Supplementary Fig. 1). To minimize the influence of spatial autocorrelation in our random-forest model, we thus applied a spatially buffer-zone-based bootstrapping procedure, subsampling the training data during the grid search procedure to make sure the distance between any two data plots is always >50 km. This buffer-zone-based bootstrap subsampling was applied 200 times and, for each subsample (~2,000 observations), we ran 48 random discrete parameter sets covering the total grid space of 240 possible parameter combinations to perform the grid search. Model performance of each model was assessed using the coefficient of determination[128] based on tenfold cross-validation and, for each subsample, we retained the model with the highest $R^2$.

To create the final community wood density maps, we used an ensemble approach, whereby we averaged the global predictions from the 200 best random-forest models based on our bootstrapping procedure. By taking the average prediction across multiple models, ensemble methods minimize the influence of any single prediction, thereby stabilizing variation and minimizing bias that can otherwise arise from extrapolation or in-fit overfitting when using a single machine learning model[129]. Moreover, by quantifying the variation across these ensemble predictions, we can identify areas that have low consensus across multiple models and which thus have higher uncertainty than would otherwise be predicted by any single model. To implement this ensemble approach, the mean predicted value across the 200 best-fitting models was used as the final model prediction for each pixel and the variation coefficient across these 200 models was used to characterize intermodel consistency (paragraph on Supplementary Fig. 3).

### Model consistency and uncertainty

Our ensemble approach allowed us to obtain spatially explicit estimates of the uncertainty associated with our random-forest models of global community-wide wood density. This was done by computing the pixel-wise variation coefficient (standard deviation divided by the mean) of the 200 bootstrapped models[126] (Supplementary Fig. 3), whereby the coefficient of variation represents the uncertainty of our wood density estimates.

### Geographic variation and drivers of community wood density properties

**Variable selection.** To explore the effect magnitude and direction of the main environmental drivers of CWD across the globe, we included

variables of high ecological importance which have shown significant relationships with wood density in previous studies[9,17,84,89,130] and performed hierarchical cluster analysis to remove highly similar variables. We then tested for multicollinearity among the retained covariates, by calculating variance inflation factors (VIFs) using the R package HH 3.1-52 (ref. [131]). All VIFs of these selected variables were <5, indicating sufficient independence among predictor variables. Mean annual temperature, soil moisture, DBH, species richness, soil C:N ratio, forest age, canopy height, LAI, human modification and fire frequency were selected for the final analysis. In addition, we included two biotic variables: angiosperm ratio and biodiversity. Specifically, the angiosperm ratio represents the proportion of angiosperm individuals within the plot, which we used to account for differences in CWD between angiosperms and gymnosperms (Fig. 1b). Biodiversity is represented by richness, calculated by scaling the observed number of species to the plot size. We excluded precipitation from our analysis due to its strong collinearity with plant water availability on a global scale and the previously established weak correlation between wood density and precipitation[1].

**Variable importance.** To test the variable importance of the selected covariates, we ran linear multivariate regression and random-forest models. To control for the potential effects of spatial autocorrelation, we ran a bootstrapping procedure, subsampling the full dataset 100 times, with each subsample randomly selecting one observation per 0.25 arc degree grid. For each subsample, we quantified the variable importance of the nine selected variables based on mean decrease in accuracy values from random-forest models using the H2O R package[127]. The average values across the 100 submodels were then used to evaluate the results (Fig. 4). Similarly, for each subsample, we fitted a multivariate regression model for the ten selected variables and calculated the corresponding regression coefficients, whereby both response and predictor variables were standardized to allow for direct effect size comparison. We then aggregated the results by calculating the mean regression coefficient and standard deviation across all submodels (Fig. 4). Furthermore, we ran the same models including only data for angiosperms or gymnosperms (Fig. 4c–f). We also tested the effects of the variables on CWD within forest types using partial linear regression models (Supplementary Fig. 5). As for the global analyses, we controlled for the effect of spatial autocorrelation by running a bootstrapping procedure, subsampling the full dataset 100 times, with each subsample randomly selecting one observation per 0.25 arc degree grid.

### Context-dependency of human disturbance and fire frequency effects

To explore the effects of human disturbances and fire frequency on CWD under different environmental conditions, we ran recursive partitioning analyses using the packages partykit[132] and ggparty[133]. We used a decision tree algorithm to explore the context-dependency of the slope and intercept of a univariate linear model for the effect of disturbance variables on community wood density properties, whereby the top four covariates based on a random-forest model (Fig. 4a) were evaluated as potential splitting points (Supplementary Fig. 6). The minimum node size (minimum number of observations contained in each terminal node) was set to 500 (~3% of the data) and the significance level was set to 0.01.

### Estimation of the living biomass in global forest

To generate a global map of aboveground tree biomass, we combined our wood density map with an existing map of live tree volume[63]. The live tree volume represents the total volume of all living trees with a DBH >10 cm, measured over bark from ground or stump height to a top stem diameter of 0 cm (ref. [63]).

$$TGB = \sum_{i=1}^{n} CWD_{mean} \times GSV \times BEFs \times \left(\frac{1}{1-RMF}\right) \qquad (2)$$

Total living tree biomass (TGB) was then calculated by multiplying our CWD estimates with live tree volume (GSV)[63]. To match the resolution of our wood density map, the GSV map was aggregated from ~100 m to ~1 km resolution. GSV represents the volume of all living trees with a diameter greater than 10 cm at breast height, measured from the ground or stump height to a top stem diameter of 0 cm, including the bark. We then used biomass expansion factors (BEFs) from the literature[75,134] (Supplementary Table 1) to convert stem biomass into aboveground tree biomass, representing the biomass of tree stems, branches, foliage, flowers and seeds[63]. Root mass fraction (RMF) is the relative proportion of plant biomass distributed to roots[65].

### Reporting summary

Further information on research design is available in the Nature Portfolio Reporting Summary linked to this article.

### Data availability

Data are available via Zenodo at https://doi.org/10.5281/zenodo.13331493 (ref. [135]).

### Code availability

Code is available via GitHub at https://github.com/LidongMo/GlobalWoodDensityProject.

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

## Acknowledgements

This work was supported by grants to L.M. from the China Scholarship Council, to C.M.Z. from the SNF Ambizione Fellowship programme (no. PZ00P3_193646) and to T.W.C. from DOB Ecology and the Bernina Foundation. We thank RESTOR (www.restor.eco) for providing data and Google Earth Engine for analytical support. This study was in part supported by the ESA CCI Biomass project funded by the European Space Agency (4000123662/18/I-NB) and The Open Earth Monitor Project funded by the European Union. The GEO-TREES initiative (https://geo-trees.org/) contributed plot data to this study, supported by the European Space Agency, IIASA, RAINFOR, AfriTRON, ForestPlots.net, ForestGEO, Smithsonian Tropical Research Institute, TmFO, Universite de Toulouse, University of Leeds, UCL and CIRAD. The French National Forest Inventory data were downloaded by GFBi at https://inventaire-forestier.ign.fr/dataifn/; the Italian Forest Inventory data were downloaded by GFBi at https://inventarioforestale.org/. O. Bouriaud acknowledges funding from the Romanian National Council for Higher Education Funding, CNFIS, project no. CNFIS-FDI-2024-F-0155. J.-C.S. considers this work a contribution to Center for Ecological Dynamics in a Novel Biosphere (ECONOVO), funded by Danish National Research Foundation (grant no. DNRF173) and his VILLUM Investigator project 'Biodiversity Dynamics in a Changing World', funded by VILLUM FONDEN (grant no. 16549). ForestPlots.net and RAINFOR contributions led by O.L.P. were supported by multiple sources including the Royal Society (GCRF International Collaboration Award ICA\R1\180100), the European Research Council (advanced grant no. 291585), the UK Natural Environment Research Council (NE/B504630/1, NE/D010306/1, NE/G012067/1, NE/D005590/1, NE/I028122/1, NE/S011811/1) and the Gordon and Betty Moore Foundation. The exploratory plots of FunDivEUROPE were established through funding from the European Union Seventh Framework Programme FP7/2007-2013 under grant no. 265171. T.M.F. was supported by a Czech Science Foundation Standard Grant (21-06446S). We thank the FCT—Portuguese Foundation for Science and Technology, project UIDB/04033/2020 and ICNF-Instituto da Conservação da Natureza, Portugal, National Forest Inventory for support. This study used GFBi plot data originally collected in Brazil with funding by Conselho Nacional de Desenvolvimento Científico e Tecnológico (CNPq) (project 520053/1998-2). We are grateful to all the ministries and agencies from the Government of Spain that supported the collection, compilation and coordination of forest inventory data, also including the Spanish Forest Inventories. S.d.M. was supported by the Serra-Húnter fellowship provided by the Government of Catalonia (Generalitat de Catalunya). C.A. and P. Schall thank the Deutsche Forschungsgemeinschaft Priority Program 1374 Biodiversity Exploratories. We acknowledge the use of data

drawn from the Natural Forest plot data collected between January 2009 and March 2014 by the LUCAS programme for the New Zealand Ministry for the Environment. Data were sourced via the NZ National Vegetation Survey Databank. Data from T.R.F. were supported by NERC (NE/W001691/1, NE/N011570/1, NE/R017980/1). K.J.S. thanks the IBL for supporting this work by internal funds under project no. 261509 AFTER FBS—maintenance of ForBioSensing project performance indicators.

## Author contributions

L.M., T.W.C. and C.M.Z. conceived, developed and wrote the paper. L.M. performed the analysis, with assistance from C.M.Z., T.W.C., D.S.M., J.v.d.H., H.M. and L.B.-M. J.L., S.d.M., G.-J.N., P.B.R. and O.L.P. provided conceptual and/or editorial input to improve the paper. M.A., Y.C.A.Y., G.A., A.M.A.Z., B.V.A., E.A.-D., P.A.-L., L.F.A., I.A., C.A., C.A.-F., A.A.-M., L.A., V.A., G.A.A., T.R.B., R.B., O. Banki, J.G.B., M.L.B., J.-F.B., L.B., P. Birnbaum, R. Bitariho, P. Boeckx, F. Bongers, C.C.F.B., O. Bouriaud, P.H.S.B., S.B., F.Q.B., R. Brienen, E.N.B., H.B., F. Bussotti, R.C.G., R.G.C., G.C., R.C., H.Y.H.C., C. Chisholm, H.C., E.C., C. Clark, D.C., G.D.C., D.A.C., F.C., J.J.C.-R., P.M.C., J.R.C., S.D., A.L.d.G., M.D., G.D., B.De.V., I.D., J.D., A.D., N.E., B.J.E., T.J.E., A.B.F., T.M.F., T.R.F., L.V.F., L. Finér, M.F., C.F., L. Frizzera, J.G.P.G., D.G., H.B.G., D.J.H., A. Hector, A. Hemp, G.H., B.H., J.L.H., M.H., P.H., A. Hillers, E.N.H., C.H., T.I., N.I., A.M.J., B.J., V.K.J., C.A.J., T.J., I.J., V.K., K. Kartawinata, E.K., D. Kenfack, D.K.K., S.K., G.K., M.L.K., T.J.K., H.S.K., K. Kitayama, M.K., H.K., F.K., D. Kucher, D.L., M.L., S.L.L., Y.L., G.L., H.L., N.V.L., B.S.M., Y.M., E.M., B.S.M., B.H.M.-J., A.R.M., E.H.M., J.K.M., J.A.M., O.M.-C., C. Mendoza, I.M.-P., S.M., C. Merow, A.M.M., V.S.M., S.A.M., P.M., M.G.N.-M., D.N., V.J.N., R.V.N., M.R.N., P.A.N., P.O., E.O.-M., Y.P., A.P., A.P.-G., E.I.P., M. Park, M. Parren, N. Parthasarathy, P.L.P., S.P., N. Picard, M.T.F.P., D.P., N.C.A.P., L.P., A.D.P., J.R.P., H.P., F.R.A., Z.R.-C., S.J.R, M.R., S.G.R., A.R., F.R., E.R., P. Saikia, C.S.-E., P. Saner, P. Schall, M.-J.S., D. Schepaschenko, M.S.-L., B.S., J.S., E.B.S., V.S., J.M.S.-D., D. Sheil, A.Z.S., A.C.DaS., J.E.S.-E., M. Silveira, J.S., P. Sist, F.S., B. Sonké, E.E.S., A.F.S., K.J.S., J.-C.S., M. Svoboda, B. Swanepoel, N. Targhetta, N. Tchebakova, H.t.S., R.T., E.T., P.M.U., V.A.U., R.V., F.V., P.M.V.B., F.v.d.P., T.V.D., M.E.v.N., R.M.V., H. Verbeeck, H. Viana, A.C.V., S.V., K.v.G., H.-F.W., J.V.W., G.D.A.W., F.W., H. Woell, V.W., R.Z., T.Z.-N., C.Z., X.Z., M.Z., Z.-X.Z. and I.C.Z.-B. provided data for the analysis. All authors reviewed and approved the paper.

## Funding

## Competing interests

The authors declare no competing interests.

## Additional information

**Correspondence and requests for materials** should be addressed to Lidong Mo.

Lidong Mo[1]✉, Thomas W. Crowther[1], Daniel S. Maynard[1,2], Johan van den Hoogen[1], Haozhi Ma[1], Lalasia Bialic-Murphy[1], Jingjing Liang[3], Sergio de-Miguel[4,5], Gert-Jan Nabuurs[6], Peter B. Reich[7,8,9], Oliver L. Phillips[10], Meinrad Abegg[11], Yves C. Adou Yao[12], Giorgio Alberti[13,14], Angelica M. Almeyda Zambrano[15], Braulio Vilchez Alvarado[16], Esteban Alvarez-Dávila[17], Patricia Alvarez-Loayza[18], Luciana F. Alves[19], Iêda Amaral[20], Christian Ammer[21], Clara Antón-Fernández[22], Alejandro Araujo-Murakami[23], Luzmila Arroyo[23], Valerio Avitabile[24], Gerardo A. Aymard[25,26], Timothy R. Baker[10], Radomir Bałazy[27], Olaf Banki[28], Jorcely G. Barroso[29], Meredith L. Bastian[30,31], Jean-Francois Bastin[32], Luca Birigazzi[33], Philippe Birnbaum[34,35,36], Robert Bitariho[37], Pascal Boeckx[38], Frans Bongers[6], Coline C. F. Boonman[39,40], Olivier Bouriaud[41], Pedro H. S. Brancalion[42], Susanne Brandl[43], Francis Q. Brearley[44], Roel Brienen[10], Eben N. Broadbent[15], Helge Bruelheide[45,46], Filippo Bussotti[47], Roberto Cazzolla Gatti[48], Ricardo G. César[42], Goran Cesljar[49], Robin Chazdon[50,51], Han Y. H. Chen[52], Chelsea Chisholm[1], Hyunkook Cho[53], Emil Cienciala[54,55], Connie Clark[56], David Clark[57], Gabriel D. Colletta[58], David A. Coomes[59], Fernando Cornejo Valverde[60], José J. Corral-Rivas[61], Philip M. Crim[62,63], Jonathan R. Cumming[62], Selvadurai Dayanandan[64], André L. de Gasper[65], Mathieu Decuyper[6], Géraldine Derroire[66], Ben DeVries[67], Ilija Djordjevic[68], Jiri Dolezal[69,70], Aurélie Dourdain[66], Nestor Laurier Engone Obiang[71], Brian J. Enquist[72,73], Teresa J. Eyre[74], Adandé Belarmain Fandohan[75], Tom M. Fayle[76,77], Ted R. Feldpausch[78], Leandro V. Ferreira[79], Leena Finér[80], Markus Fischer[81], Christine Fletcher[82], Lorenzo Frizzera[83], Javier G. P. Gamarra[84], Damiano Gianelle[83], Henry B. Glick[85], David J. Harris[86], Andrew Hector[87], Andreas Hemp[88], Geerten Hengeveld[6], Bruno Hérault[89,90], John L. Herbohn[91], Martin Herold[92], Peter Hietz[93], Annika Hillers[94,95], Eurídice N. Honorio Coronado[96], Cang Hui[97,98], Thomas Ibanez[99], Nobuo Imai[100], Andrzej M. Jagodziński[101,102], Bogdan Jaroszewicz[103], Vivian Kvist Johannsen[104], Carlos A. Joly[105], Tommaso Jucker[106], Ilbin Jung[53], Viktor Karminov[107], Kuswata Kartawinata[108], Elizabeth Kearsley[109], David Kenfack[110],

Deborah K. Kennard[111], Sebastian Kepfer-Rojas[104], Gunnar Keppel[112], Mohammed Latif Khan[113], Timothy J. Killeen[23], Hyun Seok Kim[114,115,116,117], Kanehiro Kitayama[118], Michael Köhl[119], Henn Korjus[120], Florian Kraxner[121], Dmitry Kucher[122], Diana Laarmann[120], Mait Lang[120], Simon L. Lewis[10,123], Yuanzhi Li[124], Gabriela Lopez-Gonzalez[10], Huicui Lu[125], Natalia V. Lukina[126], Brian S. Maitner[72], Yadvinder Malhi[127], Eric Marcon[128], Beatriz Schwantes Marimon[129], Ben Hur Marimon-Junior[129], Andrew R. Marshall[91,130,131], Emanuel H. Martin[132], James K. McCarthy[133], Jorge A. Meave[134], Omar Melo-Cruz[135], Casimiro Mendoza[136], Irina Mendoza-Polo[137], Stanislaw Miscicki[138], Cory Merow[50], Abel Monteagudo Mendoza[139,140], Vanessa S. Moreno[42], Sharif A. Mukul[91,141], Philip Mundhenk[119], María Guadalupe Nava-Miranda[142,143], David Neill[144], Victor J. Neldner[74], Radovan V. Nevenic[68], Michael R. Ngugi[74], Pascal A. Niklaus[145], Petr Ontikov[107], Edgar Ortiz-Malavasi[16], Yude Pan[146], Alain Paquette[147], Alexander Parada-Gutierrez[23], Elena I. Parfenova[148], Minjee Park[3,114], Marc Parren[149], Narayanaswamy Parthasarathy[150], Pablo L. Peri[151], Sebastian Pfautsch[152], Nicolas Picard[153], Maria Teresa F. Piedade[154], Daniel Piotto[155], Nigel C. A. Pitman[18], Lourens Poorter[6], Axel Dalberg Poulsen[86], John R. Poulsen[56,156], Hans Pretzsch[157,158], Freddy Ramirez Arevalo[159], Zorayda Restrepo-Correa[160], Sarah J. Richardson[133], Mirco Rodeghiero[83,161], Samir G. Rolim[155], Anand Roopsind[162], Francesco Rovero[163,164], Ervan Rutishauser[165], Purabi Saikia[166], Christian Salas-Eljatib[167,168], Philippe Saner[169], Peter Schall[21], Mart-Jan Schelhaas[6], Dmitry Schepaschenko[170,171], Michael Scherer-Lorenzen[172], Bernhard Schmid[173], Jochen Schöngart[154], Eric B. Searle[147], Vladimír Seben[174], Josep M. Serra-Diaz[175,176], Douglas Sheil[149,177], Anatoly Z. Shvidenko[121], Ana Carolina Da Silva[178], Javier E. Silva-Espejo[179], Marcos Silveira[180], James Singh[181], Plinio Sist[89], Ferry Slik[182], Bonaventure Sonké[183], Enio Egon Sosinski Jr.[184], Alexandre F. Souza[185], Krzysztof J. Stereńczak[27], Jens-Christian Svenning[40,186], Miroslav Svoboda[187], Ben Swanepoel[188], Natalia Targhetta[154], Nadja Tchebakova[148], Hans ter Steege[28,189], Raquel Thomas[190], Elena Tikhonova[126], Peter M. Umunay[191], Vladimir A. Usoltsev[192], Renato Valencia[193], Fernando Valladares[194], Peter M. Van Bodegom[195], Fons van der Plas[196], Tran Van Do[197], Michael E. van Nuland[198], Rodolfo M. Vasquez[139], Hans Verbeeck[109], Helder Viana[199,200], Alexander C. Vibrans[65,201], Simone Vieira[202], Klaus von Gadow[203], Hua-Feng Wang[204], James V. Watson[205], Gijsbert D. A. Werner[206], Florian Wittmann[207], Hannsjoerg Woell[208], Verginia Wortel[209], Roderick Zagt[210], Tomasz Zawiła-Niedźwiecki[211], Chunyu Zhang[212], Xiuhai Zhao[212], Mo Zhou[3], Zhi-Xin Zhu[204], Irie C. Zo-Bi[90] & Constantin M. Zohner[1]

[1]Institute of Integrative Biology, ETH Zurich (Swiss Federal Institute of Technology), Zurich, Switzerland. [2]Department of Genetics, Evolution and Environment, University College London, London, UK. [3]Department of Forestry and Natural Resources, Purdue University, West Lafayette, IN, USA. [4]Department of Agricultural and Forest Sciences and Engineering, University of Lleida, Lleida, Spain. [5]Forest Science and Technology Centre of Catalonia (CTFC), Solsona, Spain. [6]Wageningen University and Research, Wageningen, the Netherlands. [7]Department of Forest Resources, University of Minnesota, St. Paul, MN, USA. [8]Hawkesbury Institute for the Environment, Western Sydney University, Penrith, New South Wales, Australia. [9]Institute for Global Change Biology and School for Environment and Sustainability, University of Michigan, Ann Arbor, MI, USA. [10]School of Geography, University of Leeds, Leeds, UK. [11]Swiss Federal Institute for Forest, Snow and Landscape Research WSL, Birmensdorf, Switzerland. [12]UFR Biosciences, University Félix Houphouët-Boigny, Abidjan, Côte d'Ivoire. [13]Department of Agricultural, Food, Environmental and Animal Sciences, University of Udine, Udine, Italy. [14]National Biodiversity Future Center (NBFC), Palermo, Italy. [15]Spatial Ecology and Conservation Lab, School of Forest, Fisheries and Geomatics Sciences, University of Florida, Gainesville, FL, USA. [16]Forestry School, Tecnológico de Costa Rica TEC, Cartago, Costa Rica. [17]Fundacion Con Vida, Universidad Nacional Abierta y a Distancia (UNAD), Medellin, Colombia. [18]Field Museum of Natural History, Chicago, IL, USA. [19]Center for Tropical Research, Institute of the Environment and Sustainability, UCLA, Los Angeles, CA, USA. [20]National Institute of Amazonian Research, Manaus, Brazil. [21]Silviculture and Forest Ecology of the Temperate Zones, University of Göttingen, Göttingen, Germany. [22]Division of Forest and Forest Resources, Norwegian Institute of Bioeconomy Research (NIBIO), Ås, Norway. [23]Museo de Historia natural Noel kempff Mercado, Santa Cruz, Bolivia. [24]European Commission, Joint Research Center, Ispra, Italy. [25]UNELLEZ-Guanare, Programa de Ciencias del Agro y el Mar, Herbario Universitario (PORT), Guanare, Venezuela. [26]Compensation International S. A. Ci Progress-GreenLife, Bogotá, Colombia. [27]Department of Geomatics, Forest Research Institute, Sękocin Stary, Poland. [28]Naturalis Biodiversity Center, Leiden, the Netherlands. [29]Centro Multidisciplinar, Universidade Federal do Acre, Rio Branco, Brazil. [30]Proceedings of the National Academy of Sciences, Washington, DC, USA. [31]Department of Evolutionary Anthropology, Duke University, Durham, NC, USA. [32]TERRA Teach and Research Centre, Gembloux Agro Bio-Tech, University of Liege, Liege, Belgium. [33]Forestry Consultant, Grosseto, Italy. [34]Institut Agronomique néo-Calédonien (IAC), Nouméa, New Caledonia. [35]AMAP, Univ Montpellier, Montpellier, France. [36]CIRAD, CNRS, INRAE, IRD, Montpellier, France. [37]Institute of Tropical Forest Conservation, Mbarara University of Sciences and Technology, Mbarara, Uganda. [38]Isotope Bioscience Laboratory—ISOFYS, Ghent University, Ghent, Belgium. [39]Department of Aquatic Ecology and Environmental Biology, Institute for Water and Wetland Research, Radboud University, Nijmegen, the Netherlands. [40]Center for Ecological Dynamics in a Novel Biosphere (ECONOVO) and Center for Biodiversity Dynamics in a Changing World (BIOCHANGE), Department of Biology, Aarhus University, Aarhus, Denmark. [41]Ştefan cel Mare, University of Suceava, Suceava, Romania. [42]Department of Forest Sciences, Luiz de Queiroz College of Agriculture, University of São Paulo, Piracicaba, Brazil. [43]Bavarian State Institute of Forestry, Freising, Germany. [44]Department of Natural Sciences, Manchester Metropolitan University, Manchester, UK. [45]Institute of Biology, Geobotany and Botanical Garden, Martin Luther University Halle-Wittenberg, Halle-Wittenberg, Germany. [46]Centre for Integrative Biodiversity Research (iDiv) Halle-Jena-Leipzig, Leipzig, Germany. [47]Department of Agriculture, Food, Environment and Forest (DAGRI), University of Firenze, Florence, Italy. [48]Department of Biological, Geological and Environmental Sciences, University of Bologna, Bologna, Italy. [49]Department of Spatial Regulation, GIS and Forest Policy, Institute of Forestry, Belgrade, Serbia. [50]Department of Ecology and Evolutionary Biology, University of Connecticut, Storrs, CT, USA. [51]Tropical Forests and People Research Centre, University of the Sunshine Coast, Sippy Downs, Queensland, Australia. [52]Faculty of Natural Resources Management, Lakehead University, Thunder Bay, Ontario, Canada. [53]Division of Forest Resources Information, Korea Forest Promotion Institute, Seoul, South Korea. [54]IFER - Institute of Forest Ecosystem Research, Jilove u Prahy, Czech Republic. [55]Global Change Research Institute CAS, Brno, Czech Republic. [56]Nicholas

School of the Environment, Duke University, Durham, NC, USA. [57]Department of Biology, University of Missouri-St Louis, St. Louis, MO, USA. [58]Programa de Pós-graduação em Biologia Vegetal, Instituto de Biologia, Universidade Estadual de Campinas, Campinas, Brazil. [59]Department of Plant Sciences and Conservation Research Institute, University of Cambridge, Cambridge, UK. [60]Andes to Amazon Biodiversity Program, Madre de Dios, Peru. [61]Facultad de Ciencias Forestales y Ambientales, Universidad Juárez del Estado de Durango, Durango, Mexico. [62]Department of Biology, West Virginia University, Morgantown, WV, USA. [63]Department of Physical and Biological Sciences, The College of Saint Rose, Albany, NY, USA. [64]Biology Department, Centre for Structural and Functional Genomics, Concordia University, Montreal, Quebec, Canada. [65]Natural Science Department, Universidade Regional de Blumenau, Blumenau, Brazil. [66]Cirad, UMR EcoFoG (AgroParisTech, CNRS, INRAE, Université des Antilles, Université de la Guyane), Campus Agronomique, Kourou, French Guiana. [67]Department of Geography, Environment and Geomatics, University of Guelph, Guelph, Ontario, Canada. [68]Institute of Forestry, Belgrade, Serbia. [69]Institute of Botany, The Czech Academy of Sciences, Třeboň, Czech Republic. [70]Department of Botany, Faculty of Science, University of South Bohemia, České Budějovice, Czech Republic. [71]IRET, Herbier National du Gabon (CENAREST), Libreville, Gabon. [72]Department of Ecology and Evolutionary Biology, University of Arizona, Tucson, AZ, USA. [73]The Santa Fe Institute, Santa Fe, NM, USA. [74]Queensland Herbarium and Biodiversity Science, Department of Environment and Science, Toowong, Queensland, Australia. [75]Ecole de Foresterie et Ingénierie du Bois, Université Nationale d'Agriculture, Kétou, Benin. [76]School of Biological and Behavioural Sciences, Queen Mary University of London, London, UK. [77]Biology Centre of the Czech Academy of Sciences, Institute of Entomology, Ceske Budejovice, Czech Republic. [78]Geography, Faculty of Environment, Science and Economy, University of Exeter, Exeter, UK. [79]Museu Paraense Emílio Goeldi, Coordenação de Ciências da Terra e Ecologia, Belém, Brazil. [80]Natural Resources Institute Finland (Luke), Joensuu, Finland. [81]Institute of Plant Sciences, University of Bern, Bern, Switzerland. [82]Forest Research Institute Malaysia, Kuala Lumpur, Malaysia. [83]Research and Innovation Center, Fondazione Edmund Mach, San Michele All'adige, Italy. [84]Forestry Division, Food and Agriculture Organization of the United Nations, Rome, Italy. [85]Glick Designs LLC, Hadley, MA, USA. [86]Royal Botanic Garden Edinburgh, Edinburgh, UK. [87]Department of Biology, University of Oxford, Oxford, UK. [88]Department of Plant Systematics, University of Bayreuth, Bayreuth, Germany. [89]Cirad, UPR Forêts et Sociétés, University of Montpellier, Montpellier, France. [90]Department of Forestry and Environment, National Polytechnic Institute (INP-HB), Yamoussoukro, Côte d'Ivoire. [91]Forest Research Institute, University of the Sunshine Coast, Sippy Downs, Queensland, Australia. [92]Helmholtz GFZ German Research Centre for Geosciences, Remote Sensing and Geoinformatics Section, Telegrafenberg, Potsdam, Germany. [93]Institute of Botany, Department of Integrative Biology and Biodiversity Research, University of Natural Resources and Life Sciences Vienna, Vienna, Austria. [94]Centre for Conservation Science, The Royal Society for the Protection of Birds, Sandy, UK. [95]Wild Chimpanzee Foundation, Liberia Office, Monrovia, Liberia. [96]Instituto de Investigaciones de la Amazonía Peruana, Iquitos, Peru. [97]Centre for Invasion Biology, Department of Mathematical Sciences, National Institute for Theoretical and Computational Sciences, Stellenbosch University, Stellenbosch, South Africa. [98]Theoretical Ecology Unit, African Institute for Mathematical Sciences, Cape Town, South Africa. [99]AMAP, Univ Montpellier, CIRAD, CNRS, INRAE, IRD, Montpellier, France. [100]Department of Forest Science, Tokyo University of Agriculture, Tokyo, Japan. [101]Institute of Dendrology, Polish Academy of Sciences, Kórnik, Poland. [102]Department of Game Management and Forest Protection, Poznań University of Life Sciences, Poznań, Poland. [103]Faculty of Biology, Białowieża Geobotanical Station, University of Warsaw, Białowieża, Poland. [104]Department of Geosciences and Natural Resource Management, University of Copenhagen, Copenhagen, Denmark. [105]Department of Plant Biology, Institute of Biology, University of Campinas (UNICAMP), Campinas, Brazil. [106]School of Biological Sciences, University of Bristol, Bristol, UK. [107]Forestry Faculty, Mytischi Branch of Bauman Moscow State Technical University, Mytischi, Russian Federation. [108]Negaunee Integrative Research Center, Field Museum of Natural History, Chicago, IL, USA. [109]CAVElab-Computational and Applied Vegetation Ecology, Department of Environment, Ghent University, Ghent, Belgium. [110]CTFS-ForestGEO, Smithsonian Tropical Research Institute, Panama City, Panama. [111]Department of Physical and Environmental Sciences, Colorado Mesa University, Grand Junction, CO, USA. [112]UniSA STEM and Future Industries Institute, University of South Australia, Adelaide, South Australia, Australia. [113]Department of Botany, Dr Harisingh Gour Vishwavidyalaya (A Central University), Sagar, India. [114]Department of Agriculture, Forestry and Bioresources, Seoul National University, Seoul, South Korea. [115]Interdisciplinary Program in Agricultural and Forest Meteorology, Seoul National University, Seoul, South Korea. [116]National Center for Agro Meteorology, Seoul, South Korea. [117]Research Institute for Agriculture and Life Sciences, Seoul National University, Seoul, South Korea. [118]Graduate School of Agriculture, Kyoto University, Kyoto, Japan. [119]Institute for World Forestry, University of Hamburg, Hamburg, Germany. [120]Institute of Forestry and Engineering, Estonian University of Life Sciences, Tartu, Estonia. [121]Biodiversity and Natural Resources Program, International Institute for Applied Systems Analysis, Laxenburg, Austria. [122]Peoples Friendship University of Russia (RUDN University), Moscow, Russian Federation. [123]Department of Geography, University College London, London, UK. [124]Department of Ecology, State Key Laboratory of Biocontrol, School of Life Sciences, Sun Yat-sen University, Guangzhou, China. [125]Faculty of Forestry, Qingdao Agricultural University, Qingdao, China. [126]Center for Forest Ecology and Productivity, Russian Academy of Sciences, Moscow, Russian Federation. [127]Environmental Change Institute, School of Geography and the Environment, Oxford, UK. [128]AgroParisTech, UMR-AMAP, Cirad, CNRS, INRA, IRD, Université de Montpellier, Montpellier, France. [129]Departamento de Ciências Biológicas, Universidade do Estado de Mato Grosso, Nova Xavantina, Brazil. [130]Department of Environment and Geography, University of York, York, UK. [131]Flamingo Land Ltd, Kirby Misperton, UK. [132]Department of Wildlife Management, College of African Wildlife Management, Mweka, Tanzania. [133]Manaaki Whenua – Landcare Research, Lincoln, New Zealand. [134]Departamento de Ecología y Recursos Naturales, Facultad de Ciencias, Universidad Nacional Autónoma de México, Mexico City, Mexico. [135]Universidad del Tolima, Ibagué, Colombia. [136]Colegio de Profesionales Forestales de Cochabamba, Cochabamba, Bolivia. [137]Jardín Botánico de Medellín, Medellin, Colombia. [138]Department of Forest Management, Dendrometry and Forest Economics, Warsaw University of Life Sciences, Warsaw, Poland. [139]Jardín Botánico de Missouri, Oxapampa, Peru. [140]Universidad Nacional de San Antonio Abad del Cusco, Cusco, Peru. [141]Department of Environment and Development Studies, United International University, Dhaka, Bangladesh. [142]Instituto de Silvicultura e Industria de la Madera, Universidad Juárez del Estado de Durango, Durango, Mexico. [143]Programa de doctorado en Ingeniería para el desarrollo rural y civil, Escuela de Doctorado Internacional de la Universidad de Santiago de Compostela (EDIUS), Santiago de Compostela, Spain. [144]Universidad Estatal Amazónica, Puyo, Ecuador. [145]Department of Evolutionary Biology and Environmental Studies, University of Zürich, Zurich, Switzerland. [146]Climate, Fire and Carbon Cycle Sciences, USDA Forest Service, Durham, NC, USA. [147]Centre for Forest Research, Université du Québec à Montréal, Montréal, Québec, Canada. [148]V. N. Sukachev Institute of Forest, FRC KSC, Siberian Branch of the Russian Academy of Sciences, Krasnoyarsk, Russian Federation. [149]Forest Ecology and Forest Management Group, Wageningen University & Research, Wageningen, the Netherlands. [150]Department of Ecology and Environmental Sciences, Pondicherry University, Puducherry, India. [151]Instituto Nacional de Tecnología Agropecuaria (INTA), Universidad Nacional de la Patagonia Austral (UNPA), Consejo Nacional de Investigaciones Científicas y Técnicas (CONICET), Río Gallegos, Argentina. [152]School of Social Sciences (Urban Studies), Western Sydney University, Penrith, New South Wales, Australia. [153]GIP Ecofor, Paris, France. [154]Instituto Nacional de Pesquisas da Amazônia, Manaus, Brazil. [155]Laboratório de Dendrologia e Silvicultura Tropical, Centro de Formação em Ciências Agroflorestais, Universidade Federal do Sul da Bahia, Itabuna, Brazil. [156]The Nature Conservancy, Boulder, CO, USA. [157]Chair of Forest Growth and Yield Science, Department of Life Science Systems, TUM School of Life Sciences, Technical University of Munich, Freising, Germany. [158]Sustainable Forest Management Research Institute iuFOR, University Valladolid, Valladolid, Spain. [159]Universidad Nacional de la Amazonía Peruana,

Iquitos, Peru. [160]Servicios Ecosistémicos y Cambio Climático (SECC), Fundación Con Vida & Corporación COL-TREE, Medellín, Colombia. [161]Centro Agricoltura, Alimenti, Ambiente, University of Trento, San Michele All'adige, Italy. [162]Center for Natural Climate Solutions, Conservation International, Arlington, TX, USA. [163]Department of Biology, University of Florence, Florence, Italy. [164]Tropical Biodiversity, MUSE—Museo delle Scienze, Trento, Italy. [165]Info Flora, Geneva, Switzerland. [166]Department of Botany, Banaras Hindu University, Varanasi, India. [167]Departamento de Gestión Forestal y su Medio Ambiente, Universidad de Chile, Santiago, Chile. [168]Vicerrectoría de Investigación y Postgrado, Universidad de La Frontera, Temuco, Chile. [169]Rhino and Forest Fund e.V., Kehl, Germany. [170]International Institute for Applied Systems Analysis, Laxenburg, Austria. [171]Siberian Federal University, Krasnoyarsk, Russian Federation. [172]Geobotany, Faculty of Biology, University of Freiburg, Freiburg im Breisgau, Germany. [173]Department of Geography, Remote Sensing Laboratories, University of Zürich, Zurich, Switzerland. [174]National Forest Centre, Forest Research Institute Zvolen, Zvolen, Slovakia. [175]Université de Lorraine, AgroParisTech, INRAE, Silva, Nancy, France. [176]Center for Biodiversity Dynamics in a Changing World (BIOCHANGE), Department of Biology, Aarhus University, Aarhus, Denmark. [177]Faculty of Environmental Sciences and Natural Resource Management, Norwegian University of Life Sciences, Ås, Norway. [178]Santa Catarina State University, Lages, Brazil. [179]Departamento de Biología, Universidad de la Serena, La Serena, Chile. [180]Centro de Ciências Biológicas e da Natureza, Universidade Federal do Acre, Rio Branco, Brazil. [181]Guyana Forestry Commission, Georgetown, French Guiana. [182]Environmental and Life Sciences, Faculty of Science, Universiti Brunei Darussalam, Bandar Seri Begawan, Brunei Darussalam. [183]Plant Systematic and Ecology Laboratory, Department of Biology, Higher Teachers' Training College, University of Yaoundé I, Yaoundé, Cameroon. [184]Embrapa Recursos Genéticos e Biotecnologia, Brasilia, Brazil. [185]Departamento de Ecologia, Universidade Federal do Rio Grande do Norte, Natal, Brazil. [186]Section for Ecoinformatics and Biodiversity, Department of Biology, Aarhus University, Aarhus, Denmark. [187]Faculty of Forestry and Wood Sciences, Czech University of Life Sciences, Prague, Czech Republic. [188]Wildlife Conservation Society, Vientiane, Laos. [189]Quantitative Biodiversity Dynamics, Department of Biology, Utrecht University, Utrecht, the Netherlands. [190]Iwokrama International Centre for Rainforest Conservation and Development (IIC), Georgetown, French Guiana. [191]School of Forestry and Environmental Studies, Yale University, New Haven, CT, USA. [192]Botanical Garden of Ural Branch of Russian Academy of Sciences, Ural State Forest Engineering University, Yekaterinburg, Russian Federation. [193]Pontificia Universidad Católica del Ecuador, Quito, Ecuador. [194]LINCGlobal, Museo Nacional de Ciencias Naturales, CSIC, Madrid, Spain. [195]Institute of Environmental Sciences, Leiden University, Leiden, the Netherlands. [196]Plant Ecology and Nature Conservation Group, Wageningen University, Wageningen, the Netherlands. [197]Silviculture Research Institute, Vietnamese Academy of Forest Sciences, Hanoi, Vietnam. [198]Department of Biology, Stanford University, Stanford, CA, USA. [199]Agricultural High School, Polytechnic Institute of Viseu (IPV), Viseu, Portugal. [200]Centre for the Research and Technology of Agro-Environmental and Biological Sciences, CITAB, UTAD, Quinta de Prados, Vila Real, Portugal. [201]Department of Forest Engineering Universidade Regional de Blumenau, Blumenau, Brazil. [202]Environmental Studies and Research Center, University of Campinas (UNICAMP), Campinas, Brazil. [203]Department of Forest and Wood Science, University of Stellenbosch, Stellenbosch, South Africa. [204]Key Laboratory of Tropical Biological Resources, Ministry of Education, School of Life and Pharmaceutical Sciences, Hainan University, Haikou, China. [205]Division of Forestry and Natural Resources, West Virginia University, Morgantown, WV, USA. [206]Department of Zoology, University of Oxford, Oxford, UK. [207]Department of Wetland Ecology, Institute for Geography and Geoecology, Karlsruhe Institute for Technology, Karlsruhe, Germany. [208]Independent Researcher, Sommersbergseestrasse, Bad Aussee, Austria. [209]Centre for Agricultural Research in Suriname (CELOS), Paramaribo, Suriname. [210]Tropenbos International, Wageningen, the Netherlands. [211]Polish State Forests, Coordination Center for Environmental Projects, Warsaw, Poland. [212]Research Center of Forest Management Engineering of State Forestry and Grassland Administration, Beijing Forestry University, Beijing, China. ✉e-mail: lidong.mo@usys.ethz.ch

# Reporting Summary

## Statistics

For all statistical analyses, confirm that the following items are present in the figure legend, table legend, main text, or Methods section.

| n/a | Confirmed | |
|---|---|---|
| ☐ | ☒ | The exact sample size (*n*) for each experimental group/condition, given as a discrete number and unit of measurement |
| ☒ | ☐ | A statement on whether measurements were taken from distinct samples or whether the same sample was measured repeatedly |
| ☐ | ☒ | The statistical test(s) used AND whether they are one- or two-sided *Only common tests should be described solely by name; describe more complex techniques in the Methods section.* |
| ☐ | ☒ | A description of all covariates tested |
| ☐ | ☒ | A description of any assumptions or corrections, such as tests of normality and adjustment for multiple comparisons |
| ☐ | ☒ | A full description of the statistical parameters including central tendency (e.g. means) or other basic estimates (e.g. regression coefficient) AND variation (e.g. standard deviation) or associated estimates of uncertainty (e.g. confidence intervals) |
| ☒ | ☐ | For null hypothesis testing, the test statistic (e.g. *F*, *t*, *r*) with confidence intervals, effect sizes, degrees of freedom and *P* value noted *Give P values as exact values whenever suitable.* |
| ☒ | ☐ | For Bayesian analysis, information on the choice of priors and Markov chain Monte Carlo settings |
| ☒ | ☐ | For hierarchical and complex designs, identification of the appropriate level for tests and full reporting of outcomes |
| ☒ | ☐ | Estimates of effect sizes (e.g. Cohen's *d*, Pearson's *r*), indicating how they were calculated |

*Our web collection on statistics for biologists contains articles on many of the points above.*

## Software and code

Policy information about availability of computer code

| Data collection | Forest inventory plots data came from the Global Forest Biodiversity initiative (GFBi) database: https://www.gfbinitiative.org. Wood density data came from Global Wood Density Database (Chave, J. et al. Towards a worldwide wood economics spectrum. Ecol. Lett. 12, 351–366 (2009).), TRY database (Kattge, J. et al. TRY plant trait database–enhanced coverage and open access. Glob. Chang. Biol. 26, 119–188 (2020))  and other sources( Schepaschenko, D. et al. A database of forest biomass structure for Eurasia. (2017); Falster, D. S. et al. BAAD: a Biomass And Allometry Database for woody plants. (2015); Henry, M. et al. GlobAllomeTree: international platform for tree allometric equations to support volume, biomass and carbon assessment. Iforest 6, 326–330 (2013); Vieilledent, G. et al. New formula and conversion factor to compute basic wood density of tree species using a global wood technology database. Am. J. Bot. 105, 1653–1661 (2018); Zhang, S.-B., Slik, J. W. F., Zhang, J.-L. & Cao, K.-F. Spatial patterns of wood traits in China are controlled by phylogeny and the environment. Glob. Ecol. Biogeogr. 20, 241–250 (2011)). |
|---|---|
| Data analysis | Used R and Google earth engine for data analysis. The corresponding references are listed below: R Core Team (2023). _R: A Language and Environment for Statistical Computing_. R Foundation for Statistical Computing, Vienna, Austria. <https://www.R-project.org/>. Gorelick, N., Hancher, M., Dixon, M., Ilyushchenko, S., Thau, D., & Moore, R. (2017). Google Earth Engine: Planetary-scale geospatial analysis for everyone. Remote Sensing of Environment. |

For manuscripts utilizing custom algorithms or software that are central to the research but not yet described in published literature, software must be made available to editors and reviewers. We strongly encourage code deposition in a community repository (e.g. GitHub). See the Nature Portfolio guidelines for submitting code & software for further information.

# Data

Policy information about [availability of data](availability of data)

All manuscripts must include a [data availability statement](data availability statement). This statement should provide the following information, where applicable:

- Accession codes, unique identifiers, or web links for publicly available datasets
- A description of any restrictions on data availability
- For clinical datasets or third party data, please ensure that the statement adheres to our [policy](policy)

> Data and code can be freely accessed from the GitHub link provided below, following the publication of the paper: https://github.com/LidongMo/GlobalWoodDensityProject.

# Research involving human participants, their data, or biological material

Policy information about studies with [human participants or human data](human participants or human data). See also policy information about [sex, gender (identity/presentation), and sexual orientation](sex, gender (identity/presentation), and sexual orientation) and [race, ethnicity and racism](race, ethnicity and racism).

| | |
|---|---|
| Reporting on sex and gender | n/a |
| Reporting on race, ethnicity, or other socially relevant groupings | n/a |
| Population characteristics | n/a |
| Recruitment | n/a |
| Ethics oversight | n/a |

Note that full information on the approval of the study protocol must also be provided in the manuscript.

# Field-specific reporting

Please select the one below that is the best fit for your research. If you are not sure, read the appropriate sections before making your selection.

☐ Life sciences    ☐ Behavioural & social sciences    ☒ Ecological, evolutionary & environmental sciences

For a reference copy of the document with all sections, see [nature.com/documents/nr-reporting-summary-flat.pdf](nature.com/documents/nr-reporting-summary-flat.pdf)

# Ecological, evolutionary & environmental sciences study design

All studies must disclose on these points even when the disclosure is negative.

| | |
|---|---|
| Study description | Here, we paired ~1.1 million ground-sourced forest inventory plots from the GFBi database with collated species-level wood density data to explore global variation in wood density among both angiosperm and gymnosperm trees. Using this large-scale observation approach, we tested competing hypotheses about the dominant factors driving wood density variation across global forests, including temperature, water availability, species composition and disturbances. This approach allowed us to test theoretical predictions of geographic variation and to create a global model of wood density. We calculated community-wide mean wood density by weighting the wood density of each individual observed in a forest plot by its basal area. To explore responses to anthropogenic and natural disturbance gradients, we integrated our observations with global information on human disturbance and fire frequency. Finally, we estimated the total live forest biomass by integrating our CWD map with spatially-explicit data on live tree volume, root mass fraction, and biome-level biomass expansion factors. |
| Research sample | Forest inventory plot data was downloaded from Global Forest Biodiversity initiative (GFBi) database: https://www.gfbinitiative.org Wood density data came from Global Wood Density Database (Chave, J. et al. Towards a worldwide wood economics spectrum. Ecol. Lett. 12, 351–366 (2009).), TRY database (Kattge, J. et al. TRY plant trait database–enhanced coverage and open access. Glob. Chang. Biol. 26, 119–188 (2020))  and other sources( Schepaschenko, D. et al. A database of forest biomass structure for Eurasia. (2017); Falster, D. S. et al. BAAD: a Biomass And Allometry Database for woody plants. (2015); Henry, M. et al. GlobAllomeTree: international platform for tree allometric equations to support volume, biomass and carbon assessment. Iforest 6, 326–330 (2013); Vieilledent, G. et al. New formula and conversion factor to compute basic wood density of tree species using a global wood technology database. Am. J. Bot. 105, 1653–1661 (2018); Zhang, S.-B., Slik, J. W. F., Zhang, J.-L. & Cao, K.-F. Spatial patterns of wood traits in China are controlled by phylogeny and the environment. Glob. Ecol. Biogeogr. 20, 241–250 (2011)). |
| Sampling strategy | n/a |
| Data collection | Forest inventory plot data was downloaded from Global Forest Biodiversity initiative (GFBi) database: https://www.gfbinitiative.org Wood density data came from Global Wood Density Database (Chave, J. et al. Towards a worldwide wood economics spectrum. Ecol. Lett. 12, 351–366 (2009).), TRY database (Kattge, J. et al. TRY plant trait database–enhanced coverage and open access. Glob. Chang. |

> Biol. 26, 119–188 (2020)) and other sources( Schepaschenko, D. et al. A database of forest biomass structure for Eurasia. (2017); Falster, D. S. et al. BAAD: a Biomass And Allometry Database for woody plants. (2015); Henry, M. et al. GlobAllomeTree: international platform for tree allometric equations to support volume, biomass and carbon assessment. Iforest 6, 326–330 (2013); Vieilledent, G. et al. New formula and conversion factor to compute basic wood density of tree species using a global wood technology database. Am. J. Bot. 105, 1653–1661 (2018); Zhang, S.-B., Slik, J. W. F., Zhang, J.-L. & Cao, K.-F. Spatial patterns of wood traits in China are controlled by phylogeny and the environment. Glob. Ecol. Biogeogr. 20, 241–250 (2011)).

| | |
|---|---|
| Timing and spatial scale | The estimates of wood density and environmental covariates are represented at approximately a 1km resolution. |
| Data exclusions | n/a |
| Reproducibility | Data and code can be freely accessed from the GitHub link provided below, following the publication of the paper: https://github.com/LidongMo/GlobalWoodDensityProject. |
| Randomization | n/a |
| Blinding | n/a |

**Did the study involve field work?** ☐ Yes  ☒ No

# Reporting for specific materials, systems and methods

We require information from authors about some types of materials, experimental systems and methods used in many studies. Here, indicate whether each material, system or method listed is relevant to your study. If you are not sure if a list item applies to your research, read the appropriate section before selecting a response.

## Materials & experimental systems

| n/a | Involved in the study |
|---|---|
| ☒ | ☐ Antibodies |
| ☒ | ☐ Eukaryotic cell lines |
| ☒ | ☐ Palaeontology and archaeology |
| ☒ | ☐ Animals and other organisms |
| ☒ | ☐ Clinical data |
| ☒ | ☐ Dual use research of concern |
| ☐ | ☒ Plants |

## Methods

| n/a | Involved in the study |
|---|---|
| ☒ | ☐ ChIP-seq |
| ☒ | ☐ Flow cytometry |
| ☒ | ☐ MRI-based neuroimaging |

## Dual use research of concern

Policy information about dual use research of concern

### Hazards

Could the accidental, deliberate or reckless misuse of agents or technologies generated in the work, or the application of information presented in the manuscript, pose a threat to:

| No | Yes | |
|---|---|---|
| ☒ | ☐ | Public health |
| ☒ | ☐ | National security |
| ☒ | ☐ | Crops and/or livestock |
| ☒ | ☐ | Ecosystems |
| ☒ | ☐ | Any other significant area |

## Experiments of concern

Does the work involve any of these experiments of concern:

| No | Yes | |
|----|-----|---|
| ☒ | ☐ | Demonstrate how to render a vaccine ineffective |
| ☒ | ☐ | Confer resistance to therapeutically useful antibiotics or antiviral agents |
| ☒ | ☐ | Enhance the virulence of a pathogen or render a nonpathogen virulent |
| ☒ | ☐ | Increase transmissibility of a pathogen |
| ☒ | ☐ | Alter the host range of a pathogen |
| ☒ | ☐ | Enable evasion of diagnostic/detection modalities |
| ☒ | ☐ | Enable the weaponization of a biological agent or toxin |
| ☒ | ☐ | Any other potentially harmful combination of experiments and agents |

## Plants

| | |
|---|---|
| Seed stocks | n/a |
| Novel plant genotypes | n/a |
| Authentication | n/a |

