## [Peer Review File · Nature Ecology & Evolution]

Peer Review Information

Journal: Nature Ecology & Evolution

Manuscript Title: The global distribution and drivers of wood density and their impact on forest carbon stocks

Corresponding author name(s): Lidong Mo

Editorial Notes:

Reviewer Comments & Decisions:

Decision Letter, initial version:

21st March 2024

Dear Dr Mo,

Your manuscript entitled "Consistent climatic controls of global wood density among angiosperms and gymnosperms" has now been seen by 2 reviewers, whose comments are attached. The reviewers have raised a number of concerns which will need to be addressed before we can offer publication in Nature Ecology & Evolution. We will therefore need to see your responses to the criticisms raised and to some editorial concerns, along with a revised manuscript, before we can reach a final decision regarding publication.

We therefore invite you to revise your manuscript taking into account all reviewer and editor comments. Please highlight all changes in the manuscript text file.

* If you have not done so already please begin to revise your manuscript so that it conforms to our Article format instructions at <http://www.nature.com/natecolevol/info/final-submission>. Refer also to any guidelines provided in this letter.

2* Include a revised version of any required reporting checklist. It will be available to referees (and, potentially, statisticians) to aid in their evaluation if the manuscript goes back for peer review. A revised checklist is essential for re-review of the paper.

[REDACTED]

Nature Ecology & Evolution is committed to improving transparency in authorship. As part of our efforts in this direction, we are now requesting that all authors identified as ‘corresponding author’ on published papers create and link their Open Researcher and Contributor Identifier (ORCID) with their account on the Manuscript Tracking System (MTS), prior to acceptance. ORCID helps the scientific community achieve unambiguous attribution of all scholarly contributions. You can create and link your ORCID from the home page of the MTS by clicking on ‘Modify my Springer Nature account’. For more information please visit www.springernature.com/orcid.

Yours sincerely,

[REDACTED]

Reviewers' comments:

Reviewer #1 (Remarks to the Author):

see attached file

Reviewer #2 (Remarks to the Author):

The authors use a large data set on wood density to access the global wood density distribution and its drivers, especially for angiosperms and gymnosperms; and also estimate the total live forest biomass by integrating the wood density map with live tree volume, biomass expansion factors and root mass fractions. They map the geographic variation of wood density and show that that mean annual temperature is the most influential factor on CWD for both angiosperms and gymnosperms. Additionally, the results highlight the effects of human modification and fire risk on global forest biomass. Generally, the topic is very interesting and the manuscript is well written. However, there are some issues need to be addressed.

Lines 406-407, the wood density in tropical dry forests being up to twice as dense as that in boreal forests, which is supported by the results? This conclusion was not found in the results, and it is better to provide in the text or supporting information.

Line 408, annual temperature and soil moisture should be clear for readers.

Line 434, the 9 reference is better to be cited here.

Lines 437-444, references are needed here.

Lines 446-448, this is for gymnosperms that should be mentioned again.

Line 471, the general information of both angiosperm and gymnosperm trees is helpful for readers, such as the ranges of DBH or height, because plant size also clearly affects wood density. If possible, please provide these data.

Line 472, what hypotheses?

Line 489, maybe 32% is wrong, please to check.

Line 484, the authors focus on angiosperm and gymnosperm trees, which was indicated by the title; but authors did not show the values in figure, but show these values in lines 508-509, why? Personally, the values of angiosperm and gymnosperm trees should be shown in the first section, i.e., line 484, maybe it is more appropriate to integrate into Figure 1.

Line 494, need to check the unit.

Line 495, the full name of WWF in Table S3 should be provided.

3Line 507, conifers or gymnosperm?

Lines 515-518, g/cm³ or g cm⁻³? should be unify throughout the paper.

Line 543, need to check.

Line 579, maybe fire intensity is more important than fire frequency, but the data for fire intensity is difficult to get.

Line 674, few contents on along successional stages were mentioned throughout the manuscript.

Line 795, To determine the density of wood, whether the sample is perennial wood or current wood, please introduce it briefly.

line 797, need to check the

Line 1024, here DBH>10 cm, but the woody density data is obtained by trees with DBH > 5cm in lines 834-835, whether it will affect the final results?

The 9 reference, lack of page number.

The 51, 52, 56 need to check.

*****END*****

Author Rebuttal to Initial comments

Referees' comments:

Reviewer #1 (Remarks to the Author):

In the context of forest carbon accounting, Yang et al., 2024 collect and compile species-level wood density data to analyze the variations in wood density among both angiosperms and gymnosperms across the globe forests. This impressive new data base includes 80115 individuals tree wood density records from 10703 forest tree species.

To quantify the wood density variations across the world forests, the authors assigned species-level average wood density values to individual trees measured within 1,188,771 forest inventory plots from the GFBI data base. Wood density data could be matched to 41% of the 10703 species. When wood density information was not available at the species level or if the GFBI individual was only identified to the genus-level, mean genus-level wood density values were used instead.

At the plot level, the average community wood density (CWD) was calculated by weighting each individual tree wood density by its basal area. CWD was then quantified in four broad forests categories tropical, temperate, boreal, and dryland, each including several biomes, e.g. boreal regions including two biomes, boreal forest/taiga and tundra.

A total of 62 covariates were collected for providing information on climate, topography, soil, vegetation characteristics, fire frequency, and human disturbances in order to build a spatially-explicit models that allow to interpolate CWD across the globe forests and produce a global wood density map. Finally, the authors estimated the total live forest biomass by integrating the wood density map with spatially-explicit data on live tree volume including root mass fraction, as well as biome-level biomass expansion factors.

The main finding are a pronounced latitudinal gradient, with wood in tropical dry forests being up to twice as dense as that in boreal forests. In both angiosperms and gymnosperms, temperature and water availability are the primary factors influencing the variation in wood density globally. At more local scales wood density variations result from disturbance, such as forest management and fire risk.

Finally the global tree biomass calculated with the spatially-explicit wood density model is higher by 4% than the biomass calculated with a constant wood density value of 0.53 g/cm³ (the global average). However higher differences emerged in different biomes (tropical moist-12%, tropical dry17%, tropical savanna-17% and Mediterranean forests-21%).

The authors certainly performed a huge piece of work and demonstrate high level skills in data processing and modelling. The new wood density data base and the global wood density map are two useful and impressive results.

Reply 1: We thank the reviewer for their thorough and positive assessment of our paper.

This said, I have several comments and suggestion aiming at improving your contribution

Why the authors does not cite the work authored by Yang, H et al., 2024 on the Global patterns of tree wood density? In their paper Yang et al. used almost the same wood density data bases than in this manuscript for producing a global wood density map as well. I feel

interesting and needed to evaluate the pro and cons of the approaches carried out by the two groups of authors, for the data selection, validation and the modeling approaches.

Reply 2: This is a great point. We now cite Yang et al. 2024. Although we knew of their work, we did not cite the paper because we had submitted our manuscript to Nature Ecology & Evolution approximately two weeks prior to the publication of the paper by Yang et al., 2024.

The fundamental distinction between our approach and that of Yang et al. lies in our use of forest inventory data to represent community-level wood density, while Yang et al. base their analysis on observations of wood density at the individual tree level. Upon comparing the two products, now illustrated in Figure S9, it is evident that our map displays a broader range of wood density variations. This variation may stem from outliers in the Yang et al. model, which likely led to a relatively conservative random forest model that is more centered around the mean. Yang et al. calculated the pixel-level average wood density by averaging all tree-level wood density measurements for each leaf type and habit in each grid cell ($0.01^\circ \times 0.01^\circ$), and then used these measurements as the dependent variable for their modeling process. This means that in some cases, the estimated pixel-level wood density may just stem from measurements of a single individual. In contrast, by including forest inventory data, our study calculated the community-wide wood density for each plot (CWD), weighted by tree basal area for each individual. Therefore, our wood density reflects plot-level or community-level functional traits and community characteristics. Since the spatial modeling of community-wide wood density was performed at a 30 arc-second ($\sim 1 \text{ km}^2$) resolution, we aggregated CWD values within each 30 arc-second pixels by calculating the mean.

Additionally, considering spatial autocorrelation, Yang et al. evaluated the performance of their machine learning models using a leave-one-cluster-out cross-validation method, which was performed on clusters defined by eight different methods. Regardless of the strength of the spatial autocorrelation, the predicted maps were still influenced by spatial autocorrelation to varying degrees. In our study, we introduced a spatially buffered zone-based bootstrapping procedure, subsampling the training data during the grid search procedure to ensure the distance between any two data plots always exceeded 50 km, the distance at which spatial autocorrelation effects are significant. Consequently, our final products do not introduce spatial autocorrelation, eliminating the need for any post-validation procedures regarding spatial autocorrelation.

Furthermore, Yang et al. used four models to predict wood density for different leaf types and habits, and then used a global map of plant functional type fractions from the ESA-CCI as weighting factors to calculate average wood density values for each pixel. In contrast, our study used one type of machine learning model fed by bootstrapped subsamples.

To highlight the differences between the two studies, we conducted a spatial and biome-level comparison of the wood density maps from Yang et al. with our wood density products (Figure S9 and line 711-724).

Forest inventory data: national forest inventories provide unique and necessary ground data that are not easy to compare between countries: the author do not address this question and do not explain how their Forest Inventory data were harmonized: for instance as the date of measurement of the more than one million forest plots varies (differences in decades in some cases), how original measurements are corrected for allowing the forest biomass calculation. My suggestion is to consider the paper authored by Avitabile et al., 2024 on harmonized statistics and maps of forest biomass and increment in 38 European countries.

These best possible forest biomass references could be used in this manuscript as a reference for the evaluation of the simulated biomass results. Maybe one option to go one step further than Yang et al., 2024.

Reply 3: We fully agree that harmonizing various forest inventory plots investigated in different years and locations can be challenging. To mitigate issues related to data age, we implemented a data cleaning process that excluded observations from decades past, retaining only the most recent ones. The median year of our cleaned data is 2003. Additionally, to assess the impact of temporal changes in species composition on our results, we now applied a random effects model to plots with time series information. The model, defined as Wood density \sim (1|Plot) + (1|Year), showed that variance in wood density was predominantly (97.9%) attributable to differences across plots, with only 0.2% due to variations across years (line 872-877). This demonstrates that our plot-level wood density estimates are minimally influenced by the year of observation.

Furthermore, for biomass calculations in this paper, we consistently used the growing stock volume and forest cover datasets from the year 2010. By doing so, we ensured that our biomass estimates reflect the state of global forests in that year, thereby maintaining consistency in our analysis.

Thank you for your suggestion to validate using the updated map from Avitabile et al., 2024. The Avitabile et al., 2024 map of forest biomass density was obtained from the ESA CCI Biomass map, which is also what we used here. We compared and validated our findings with the bias-corrected ESA-CCI biomass product from Araza et al., 2022, please refer to line 654-662 and Supplementary Fig. S7. The only difference is that we use the year 2010 instead of 2020 as reference year, as this better aligns with the forest inventory data.

Wood density data: the authors do not discuss the compatibility between the different wood density data bases used for their work nor the consequences of possible bias on their analysis. Sampling bias were identified, e.g. Henry et al., 2010 point out the difference in wood density measured from tree destructive testing and wood density from existing databases, while Williamson et al., 2005 point out methodological issues, eg moisture control. I tried to look inside each data base and came to the conclusion that at least the one authored by Schepaschenko et al is probably very different from the others: the original data are fresh volumes and dry weights of the whole tree components (stems, branches etc, similar to Henry et al., 2010) while in most of the other wood density data bases the measurements are carried out on small specimens (clear wood samples or increment cores). As the average tree stem wood density differs from the wood density at DBH, how to account for such bias? I also suggest to discuss the intra tree species wood density variability (limits of the approach when the nb of records/species is low, e.g. <10).

Reply 4: We fully agree that ensuring agreement between various databases recording wood density is essential. To assess this agreement, we now compared the nine data sources by examining species common to any two databases. We tested for agreement by building a linear regression model for all common species pairs and calculated the R^2 value, which was 0.78 (see Fig. S8, line 837-841). This indicates high consistency among all nine data sources and minimal bias from different wood density determination methods. Additionally, all data points are distributed around the 1:1 line, showing that there is no systematic bias. The subplots in Fig. S8 b-j compare the species pairs of each database with all the other databases and also show that all data points are well distributed around the 1:1 line.

The data from Schepaschenko et al. has the fewest common species with other databases, but these points are also close to the 1:1 line (Supplementary Fig. S8c and line 837-841), indicating consistency with other databases. Similarly, the common species between Database 6 from Henry et al. and other databases also have a distribution close to the 1:1 line, further indicating consistency.

We now also explored the intraspecific variations of the species. In addition to the random effects model described in the methods section (line 861-864), we calculated the variation coefficients for the 5,527 species that have between 3 and 10 observations, as per your suggestion. Overall, 82% of the species have variation coefficients smaller than 0.1, and 48% have coefficients smaller than 0.01 (Supplementary Fig. S10). These statistics indicate that intraspecific variation contributes minimally to the overall wood density variation. This finding is supported by the results of the random effects model (line 855-861), where we found that approximately 81% of the total variation in our wood density data is explained by taxonomic information at the family, genus and species levels, with 24% of the variation explained by family information, 30% by genus information and an additional 27% explained by species information.

Overall, the above analyses suggest that the noise inherent to large databases, such as the wood density data we used, has limited effects on the overall results and does not lead to systematic biases.

Wood density variations along climatic and environmental gradient: maybe your results could be compared to those obtained for European species by Kerfriden et al. For both angiosperms and gymnosperms, they illustrate how wood density (stem biomass ratio or CWD) decreases with temperature and soil water capacity. The altitudinal gradient, is similar to the latitudinal gradient as temperature decreases with altitude.

Reply 5: Thank you very much for pointing us toward the Kerfriden paper, we had not seen this paper yet. We now discuss their results and compare them with ours in line 595-598 and 897-900. According to the results presented in Fig. 1 of Kerfriden et al., they indeed find highly similar responses to environmental gradients.

Furthermore, in Kerfriden et al., the impact of intraspecific variation on large-scale wood density variation was tested using two indices: SBRm and SBRI. SBRm used species mean wood density, while SBRI used individual tree values. No systematic deviation was found between the two metrics across environmental gradients. This implies that using species-averaged wood density to estimate community-level wood density does not significantly affect community-level statistics, highlighting the conservative nature of wood density at the species level.

Accounting for the size of the trees for explaining the wood density variations? At the stand levels tree size/dbh is correlated with stand age and we know that for important European commercial species, wood density varies with age/DBH, decreasing trend for oaks, increasing for beech, increasing for pines as well for spruces and firs (e.g., Bouriaud et al., 2015, Franceschini et al., 2017, Zeller et al., 2017).

Reply 6: We appreciate the reviewer's suggestion to consider the effects of DBH on variations in wood density. We now included DBH in our models of the drivers of wood density. The results show that, relative to climate drivers, DBH has a minor impact on global wood density (line 608-609). However, we fully agree that, at more local scales, DBH and stand age should will play a more important role.

Wood density and wood anatomy: as wood density results directly from the wood anatomy, it is useful to classify tree forest species in four groups, conifers and three groups for hardwoods, ring porous, semiring porous, diffuse porous. Such classification could maybe enhance the discussion of your results.

Reply 7: While it is straightforward to obtain data on angio versus gymnosperms, it is unfortunately not possible to obtain porosity data for all 10,703 species in our dataset. However, as we were mostly interested in biogeographic patterns in this study, we believe such detail is beyond the scope of our research.

At last, the title of the manuscript could maybe better describe the content and value of the work, I would suggest to mention “global wood density map” or reassessment of global forest biomass.

Reply 8: Thank you for your suggestion. We now changed the tile to ‘The global distribution and drivers of wood density across angiosperms and gymnosperms and their impact on forest carbon stocks’.

Other minor’s comments

I was unable to look at your new wood density data as the file is not yet available, hopefully it should be as mentioned in line 808 (Data is available at GitHub:
<https://github.com/LidongMo/GlobalWoodDensityProject>)

Reply 9: Yes, all code and data will be made openly available once the paper is published.

Is it possible to display the forest biomass volumes per biomes versus nb of species? The question behind is the description of the sampling design, not so minor!

Reply 10: We have now added an additional analysis to display the relative relationship between the forest growing stock volumes and the number of species involved in the analysis for each biome (Supplementary Figure S11).

Maybe necessary to check the numbers. When I used the numbers given in lines 798-807, I found 77372 records from 19898 species, which is different from the 80115 individuals’ tree wood density records mentioned in line 806. I understand that the difference in the number of species result from the species names duplication/synonyms

Reply 11: Now updated. In total, there are 77,372 wood density records and 10,703 species included in the nine data sources, after correcting for species names using the TNRS package in R.

The average number of records per species is about 7, and according to the different sources, it varies between about 1 up to 50 records per species, which is better than nothing, however low especially to account for intraspecific wood density variability. This is very important for commercial forest species present along wide latitudinal gradients

Regarding the ref 51 (Schepaschenko et al.) I found more than 7000 records for 58 species

Source	Nb records	Nb species	Nb/species	Schepaschenko		
Chave & Zanne	16468	8412	1,96	1	Pinus sylvestris L.	3232
TRY	46668	7514	6,21	2	Betula alba L.	908
Brown, ref 50	1117	937	1,19	3	Picea abies (L.) Karst.	541
Vieilledent, ref 54	4022	872	4,61	4	Pinus Pallasiana	434
Zhang, ref 55	618	615	1,00	5	Picea obovata L.	316
Henry, ref 53	624	250	2,50	6	Populus tremula L.	274
Schepaschenko, ref 51	3092	58	53,31	7	Tilia cordata Mill.	226
Falster, ref 52	3529	179	19,72	8	Abies alba Mill. Mill.	200
Google scholar	1234	1061	1,16	9	Larix cajanderi Mayr.	144
	77372	19898		10	Larix sibirica L.	106
	80115	10703	7,49	11	Pinus sibirica Du Tour	75

Reply 12: While we agree that intra-specific variation is important for commercial forest species, such variation was beyond the scope of our research. As also mentioned in replies 4 and 5, intra-specific variation is unlikely to significantly affect the large-scale gradients in wood density we were interested in here. As described in our methods (line 858-861), we quantified the extent of inter-species variation in wood density by running a random-effects model on all 77,372 observations, including family, genus, and species as random effects and wood density as response variable. The model showed that ~81% of the individual variation in wood density is explained by taxonomic information on family, genus and species. While some of the remaining 19% unexplained variation may be due to intraspecific variation, the majority is accounted for without it. In addition, we calculated the variation coefficients for the 5,527 species with between 3 and 10 observations. Overall, 82% of the species have variation coefficients smaller than 0.1, and 48% have coefficients smaller than 0.01 (Supplementary Figure S10)

I share two questions with the authors, not necessary to respond
 Authorship

I was impressed by the number of co-authors, 215! In addition well recognized scientists. How is it possible to have 215 co-authors? In case of interest the authors could consider the proposals formulated by Ewers et al., 2019. They suggest to divorce authorship of a manuscript from authorship of the resources used in the manuscript, which can be achieved by creating separate categories of authorship: manuscript and resource authors.

Reply 13: It is policy of GFBi and TRY to include all data contributors as authors. Additionally, all these co-authors were involved in data preparation and, at least proof-read the manuscript. We have now clarified the contribution of each co-author in the author contribution statement.

Parachute science

I read with interest the comment by Bhaurnik, 2023 inspired from Miller et al., 2023. They point out that studies conducted in the Global South are led by scientists based in the Global North with limited involvement of local researchers. As the wood density data bases from tropical forests are often com

Reply 14: We fully agree that inclusion of scientists from the Global South is crucial. Upon reviewing our list of contributors, we have identified 80 co-authors from these regions, who represent approximately 25% of our total cohort of 236 authors. We acknowledge the necessity of increasing engagement from the Global South and are committed to advocating

for enhanced funding and more opportunities to facilitate their participation in global research initiatives. Additionally, it is important to note that, should we exclude contributors who primarily provided data, a significant proportion of researchers from the Global South would be omitted. We believe this integrative approach is highly beneficial as it broadens authorship beyond those who are most comfortable providing extensive manuscript feedback, a practice often biased towards native speakers from North America and Europe.

Reviewer #2 (Remarks to the Author):

The authors use a large data set on wood density to access the global wood density distribution and its drivers, especially for angiosperms and gymnosperms; and also estimate the total live forest biomass by integrating the wood density map with live tree volume, biomass expansion factors and root mass fractions. They map the geographic variation of wood density and show that that mean annual temperature is the most influential factor on CWD for both angiosperms and gymnosperms. Additionally, the results highlight the effects of human modification and fire risk on global forest biomass. Generally, the topic is very interesting and the manuscript is well written. However, there are some issues need to be addressed.

Reply 15: Thank you very much for the positive comments on our paper.

Lines 406-407, the wood density in tropical dry forests being up to twice as dense as that in boreal forests, which is supported by the results? This conclusion was not found in the results, and it is better to provide in the text or supporting information.

Reply 16: Thank you for detecting this oversight. The number is now corrected. See line 405-406.

Line 408, annual temperature and soil moisture should be clear for readers.

Reply 17: Rewritten. See line 408.

Line 434, the 9 reference is better to be cited here.

Reply 18: Reference 9 now cited. See line 434.

Lines 437-444, references are needed here.

Reply 19: Reference added.

Lines 446-448, this is for gymnosperms that should be mentioned again.

Reply 20: Now clarified (line 486).

Line 471, the general information of both angiosperm and gymnosperm trees is helpful for readers, such as the ranges of DBH or height, because plant size also clearly affects wood density. If possible, please provide these data.

Reply 21: We now provide the DBH ranges of gymnosperm and angiosperm trees (see line 892-894). However, height information for each of the individuals is not provided by the GFDL database.

Line 472, what hypotheses?

Reply 22: Clarified, see lines 471-474.

Line 489, maybe 32% is wrong, please to check.

Reply 23: Corrected, it was 28% (line 494).

Line 484, the authors focus on angiosperm and gymnosperm trees, which was indicated by the title; but authors did not show the values in figure, but show these values in lines 508-509, why? Personally, the values of angiosperm and gymnosperm trees should be shown in the first section, i.e., line 484, maybe it is more appropriate to integrate into Figure 1.

Reply 24: We now added the description in lines 488-489. In addition, we added the wood density comparison between gymnosperm and angiosperm trees to Fig. 1.

Line 494, need to check the unit.

Reply 25: Corrected. See also reply 28 for units in lines 520-526.

Line 495, the full name of WWF in Table S3 should be provided.

Reply 26: Done.

Line 507, conifers or gymnosperm?

Reply 27: Corrected to "gymnosperm".

Lines 515-518, g/cm³ or g cm⁻³? should be unify throughout the paper.

Reply 28: We now consistently use g/cm³.

Line 543, need to check.

Reply 29: We now rewrote this sentence to clarify the spatial subsampling. See line 547-550.

Line 579, maybe fire intensity is more important than fire frequency, but the data for fire intensity is difficult to get.

Reply 30: We absolutely agree with this. The potential for fire intensity data to improve ecological modeling is now mentioned in line 620-622.

Line 674, few contents on along successional stages were mentioned throughout the manuscript.

Reply 31: The pattern of CWD in our study results not only from environmental conditions but has also been influenced by successional stages, which we tried to capture to some degree by including forest age in the driver analysis. However, the focus of our study is on broad-scale spatial patterns and we lack direct data and observations to investigate successional effects on wood density in more detail. Nevertheless, we think it is important to mention the

potential impact of forest succession to acknowledge that wood density often changes along successional stages.

Line 795. To determine the density of wood, whether the sample is perennial wood or current wood, please introduce it briefly.

Reply 32: By “current”, does the reviewer mean the wood formed in the current year? This will normally be included in the sample, but the density of all other tree rings will also be measured.

line 797, need to check the

Reply 33: We checked and updated the numbers. line 820-836.

Line 1024, here DBH>10 cm, but the woody density data is obtained by trees with DBH > 5cm in lines 834-835, whether it will affect the final results?

Reply 34: This is a good point. In our analysis of the community wood density for each forest plot, we aimed to capture a comprehensive snapshot by including as many individuals as possible from the inventory plots, setting a minimum DBH threshold of 5 cm to represent the community broadly. However, for estimating forest biomass, we relied on satellite-derived growing stock volume data, which only detects trees with a DBH greater than 10 cm. Consequently, trees smaller than 10 cm were not included in our biomass estimates. Despite this discrepancy, the impact on our results is minimal. On average, small trees—with a DBH ranging from 5 to 10 cm—constitute less than 5% of the total basal area in each plot. Thus, their contribution to the overall community-level statistics regarding functional traits and biomass is relatively minor. This ensures that our findings remain robust despite the methodological differences between the measurements of wood density and biomass estimation.

The 9 reference, lack of page number.

Reply 35: Reformatted the reference information.

The 51, 52, 56 need to check.

Reply 36: References now corrected.

Decision Letter, first revision:

Our ref: NATECOLEVOL-24010118A

25th July 2024

Dear Dr. Mo,

Thank you for your patience as we've prepared the guidelines for final submission of your Nature Ecology & Evolution manuscript, "The global distribution and drivers of wood density across angiosperms and gymnosperms and their impact on forest carbon stocks" (NATECOLEVOL-24010118A). Please carefully follow the step-by-step instructions provided in the attached file, and add a response in each row of the table to indicate the changes that you have made. Please also check and comment on any additional marked-up edits we have proposed within the text. Ensuring that each point is addressed will help to ensure that your revised manuscript can be swiftly handed over to our production team.

****We would like to start working on your revised paper, with all of the requested files and forms, as soon as possible (preferably within two weeks). Please get in contact with us immediately if you anticipate it taking more than two weeks to submit these revised files.****

In recognition of the time and expertise our reviewers provide to Nature Ecology & Evolution's editorial process, we would like to formally acknowledge their contribution to the external peer review of your manuscript entitled "The global distribution and drivers of wood density across angiosperms and gymnosperms and their impact on forest carbon stocks". For those reviewers who give their assent, we will be publishing their names alongside the published article.

14Nature Ecology & Evolution offers a Transparent Peer Review option for new original research manuscripts submitted after December 1st, 2019. As part of this initiative, we encourage our authors to support increased transparency into the peer review process by agreeing to have the reviewer comments, author rebuttal letters, and editorial decision letters published as a Supplementary item. When you submit your final files please clearly state in your cover letter whether or not you would like to participate in this initiative. Please note that failure to state your preference will result in delays in accepting your manuscript for publication.

Cover suggestions

We welcome submissions of artwork for consideration for our cover. For more information, please see our guide for cover artwork.

Nature Ecology & Evolution has now transitioned to a unified Rights Collection system which will allow our Author Services team to quickly and easily collect the rights and permissions required to publish your work. Approximately 10 days after your paper is formally accepted, you will receive an email in providing you with a link to complete the grant of rights. If your paper is eligible for Open Access, our Author Services team will also be in touch regarding any additional information that may be required to arrange payment for your article.

Please note that *Nature Ecology & Evolution* is a Transformative Journal (TJ). Authors may publish their research with us through the traditional subscription access route or make their paper immediately open access through payment of an article-processing charge (APC). Authors will not be required to make a final decision about access to their article until it has been accepted. Find out more about Transformative Journals

Authors may need to take specific actions to achieve compliance with funder and institutional open access mandates. If your research is supported by a funder that requires immediate open access (e.g.

according to Plan S principles) then you should select the gold OA route, and we will direct you to the compliant route where possible. For authors selecting the subscription publication route, the journal's standard licensing terms will need to be accepted, including <https://www.nature.com/nature-portfolio/editorial-policies/self-archiving-and-license-to-publish>. Those licensing terms will supersede any other terms that the author or any third party may assert apply to any version of the manuscript.

[REDACTED]

[REDACTED]

Reviewer #1:

Remarks to the Author:

Dear authors,

Thank you for having submitted your revised manuscript. Your additional work and modifications took into account my comments on the initial submission

For me, it is now ok for publication

Find below four minor suggestions for hopefully improving readability

Best regards

Jean-Michel LEBAN

1-Author contribution

Replace initials with first names and names for lines 1408 to 1410, easier to read

2-Figure S9

Very informative figure, why not include it in the paper instead of the supplementary material?

3-Figure S10

Replace “within individuals” with “between individuals” or “within species wood density measurements”

4-Figure S11

In my view, this figure is better suited to the paper itself. In addition, I suggest not to log transform both axes, direct reading of numbers provides much more clearer information on the huge differences between biomes, both in terms of standing volumes and number of species per biome

Reviewer #2:

Remarks to the Author:

The revised manuscript generally addressed the issues mentioned in last version, but still some issues need to be addressed. Personally, it can be accepted after minor revision.

Line 440, need to check;

Lines 618-620, the results were not supported by fig 4ace, e.g., human modification as the forth most important factor affecting CWD across all plots and its importance increased in gymnosperm-only and angiosperm-only communities (Fig. 4c,e), which were shown in fig 4a.

Line 623, eight variables? Nine variables were mentioned in line 589.

Author Rebuttal, first revision:Referees' comments:

Reviewer #1 (Remarks to the Author):

Dear authors,

Thank you for having submitted your revised manuscript. Your additional work and modifications took into account my comments on the initial submission
For me, it is now ok for publication
Find below four minor suggestions for hopefully improving readability

Best regards
Jean-Michel LEBAN

Reply 1: Thank you so much for your valuable feedback and insightful comments, which have greatly contributed to improving the quality of our work.

1-Author contribution

Replace initials with first names and names for lines 1408 to 1410, easier to read [Ed comment: please ignore this as initials are house style for author contributions]

Reply 2: NA.

2-Figure S9

Very informative figure, why not include it in the paper instead of the supplementary material?

Reply 3: Thank you so much for your suggestion. Given that the figure is not a main result of our study, we prefer to keep this figure in the supplementary materials, but are open to move it to the main text should the editor prefer this option.

3-Figure S10

Replace “within individuals” with “between individuals” or “within species wood density measurements”

Reply 4: Changed.

4-Figure S11

In my view, this figure is better suited to the paper itself. In addition, I suggest not to log transform both axes, direct reading of numbers provides much more clearer information on the huge differences between biomes, both in terms of standing volumes and number of species per biome

Reply 5: Figure is now shown without log transformation (see Supplementary Figure S11).

Reviewer #2 (Remarks to the Author):

The revised manuscript generally addressed the issues mentioned in last version, but still some issues need to be addressed. Personally, it can be accepted after minor revision.

Reply 6: We greatly appreciate the reviewers' time and effort in reviewing our work and contribution to improving our paper.

Line 440. need to check;

Reply 7: 'vapor pressure' now changed to 'vapor pressure deficit' (Line 442).

Lines 618-620, the results were not supported by fig 4ace, e.g., human modification as the forth most important factor affecting CWD across all plots and its importance increased in gymnosperm-only and angiosperm-only communities (Fig. 4c,e), which were shown in fig 4a.

Reply 8: We do not fully understand this comment. Does the reviewer mean that the importance relative to other factors decreases in gymnosperm-only and angiosperm-only communities?

Line 623, eight variables? Nine variables were mentioned in line 589.

Reply 9: Corrected.

Final Decision Letter:

Dear Dr Mo,

We are pleased to inform you that your Article entitled "The global distribution and drivers of wood density and their impact on forest carbon stocks" has now been accepted for publication in *Nature Ecology & Evolution*.

Over the next few weeks, your paper will be copyedited to ensure that it conforms to *Nature Ecology and Evolution* style. Once your paper is typeset, you will receive an email with a link to choose the appropriate publishing options for your paper and our Author Services team will be in touch regarding any additional information that may be required

Due to the importance of these deadlines, we ask you please us know now whether you will be difficult to contact over the next month. If this is the case, we ask you provide us with the contact information (email, phone and fax) of someone who will be able to check the proofs on your behalf, and who will be available to address any last-minute problems . Once your paper has been scheduled for online publication, the Nature press office will be in touch to confirm the details.

Acceptance of your manuscript is conditional on all authors' agreement with our publication policies (see www.nature.com/authors/policies/index.html). In particular your manuscript must not be published elsewhere and there must be no announcement of the work to any media outlet until the publication date (the day on which it is uploaded onto our web site).

Please note that *Nature Ecology & Evolution* is a Transformative Journal (TJ). Authors may publish their research with us through the traditional subscription access route or make their paper immediately open access through payment of an article-processing charge (APC). Authors will not be required to make a final decision about access to their article until it has been accepted. Find out more about Transformative Journals

Authors may need to take specific actions to achieve compliance with funder and institutional open

21access mandates. If your research is supported by a funder that requires immediate open access (e.g. according to Plan S principles) then you should select the gold OA route, and we will direct you to the compliant route where possible. For authors selecting the subscription publication route, the journal's standard licensing terms will need to be accepted, including <https://www.nature.com/nature-portfolio/editorial-policies/self-archiving-and-license-to-publish>. Those licensing terms will supersede any other terms that the author or any third party may assert apply to any version of the manuscript.

We welcome the submission of potential cover material (including a short caption of around 40 words) related to your manuscript; suggestions should be sent to Nature Ecology & Evolution as electronic files (the image should be 300 dpi at 210 x 297 mm in either TIFF or JPEG format). Please note that such pictures should be selected more for their aesthetic appeal than for their scientific content, and that colour images work better than black and white or grayscale images. Please do not try to design a cover with the Nature Ecology & Evolution logo etc., and please do not submit composites of images related to your work. I am sure you will understand that we cannot make any promise as to whether any of your suggestions might be selected for the cover of the journal.

You can generate the link yourself when you receive your article DOI by entering it here: <http://authors.springernature.com/share>.

[REDACTED]

PS Click on the following link if you would like to recommend Nature Ecology & Evolution to your librarian: <http://www.nature.com/subscriptions/recommend.html#forms>

** Visit the Springer Nature Editorial and Publishing website at www.springernature.com/editorial-and-publishing-jobs for more information about our career opportunities. If you have any questions please click here.**